# Motility and tumor infiltration are key aspects of invariant natural killer T cell anti-tumor function

Chenxi Tian [1], Yu Wang[1], Miya Su[1], Yuanyuan Huang[1], Yuwei Zhang[1], Jiaxiang Dou[2], Changfeng Zhao[1], Yuting Cai[1], Jun Pan[1], Shiyu Bai[1], Qielan Wu[1], Sanwei Chen[3], Shuhang Li[1], Di Xie[1], Rong Lv[4], Yusheng Chen[2], Yucai Wang [1,5], Sicheng Fu [1] ✉, Huimin Zhang [1] ✉ & Li Bai [1,2,5,6] ✉

Dysfunction of invariant natural killer T (iNKT) cells contributes to immune resistance of tumors. Most mechanistic studies focus on their static functional status before or after activation, not considering motility as an important characteristic for antigen scanning and thus anti-tumor capability. Here we show via intravital imaging, that impaired motility of iNKT cells and their exclusion from tumors both contribute to the diminished anti-tumor iNKT cell response. Mechanistically, CD1d, expressed on macrophages, interferes with tumor infiltration of iNKT cells and iNKT-DC interactions but does not influence their intratumoral motility. VCAM1, expressed by cancer cells, restricts iNKT cell motility and inhibits their antigen scanning and activation by DCs via reducing CDC42 expression. Blocking VCAM1-CD49d signaling improves motility and activation of intratumoral iNKT cells, and consequently augments their anti-tumor function. Interference with macrophage-iNKT cell interactions further enhances the anti-tumor capability of iNKT cells. Thus, our findings provide a direction to enhance the efficacy of iNKT cell-based immunotherapy via motility regulation.

Invariant natural killer T (iNKT) cells are innate-like T cells which express semi-invariant TCR and response quickly to CD1d-presented lipid antigens[1]. They have cytotoxic function and release large amounts of Th1 and Th2 cytokines upon activation, and are able to regulate functions of dendritic cells (DCs), natural killer (NK) cells, and CD8 T cells either directly or indirectly[2–4]. Due to their direct and indirect tumor-killing effects, these cells have been reported as important players in anti-tumor immunotherapy[5,6]. The barriers limit anti-tumor efficacy of iNKT cells include immune exclusion and dysfunction of these cells in tumors[7–9]. Chemokines and adhesion molecules are known to regulate entry of immune cells into tumors[10–12], and tumor microenvironment is reported to cause impaired cell function[13–15]. Several studies have explored the mechanisms underlying dysfunction of intratumoral iNKT cells and made significant progress[7,8,16]. Particularly, the metabolic reprogramming alters signaling cascade and leads to dysfunction of iNKT cells[7,8,16]. Additionally, immunosuppressive molecules in tumors might also make contributions[17,18]. Notably, these factors mainly relate to dysfunctional status of iNKT cells before or after activation, representing a relatively static paradigm[19]. On the other hand, cell motility is essential for immune cell activation and

[1]Hefei national Research Center for Physical Sciences at the Microscale, Center for Advanced Interdisciplinary Science and Biomedicine of IHM, Division of Life Sciences and Medicine, University of Science and Technology of China, Hefei, China. [2]Institute of Health and Medicine, Hefei Comprehensive National Science Center, Hefei, China. [3]The First Affiliated Hospital of Anhui Medical University, Hefei, China. [4]Anhui Blood Center, Heifei, China. [5]Biomedical Sciences and Health Laboratory of Anhui Province, Division of Life Sciences and Medicine, University of Science and Technology of China, Hefei, China. [6]National Synchrotron Radiation Laboratory, University of Science and Technology of China, Hefei, China. ✉e-mail: fsc@mail.ustc.edu.cn; hmzhang@ustc.edu.cn; baili@ustc.edu.cn

proper immune responses in vivo[11,20–22]. It allows tumor infiltration, antigen scanning, cell interaction and activation, implying a motile regulation of immune cell function[22–26]. Indeed, reduced motility of T cells has been reported in tumors and relates to hindered anti-tumor responses[27] despite the unclear mechanisms. To date, the factors controlling motility of intratumoral iNKT cells and their contribution to impaired iNKT cell function remain to be explored.

Both tumor infiltration and intratumoral motility are closely related to adhesion molecules[28,29]. Vascular endothelial cadherin (VE-cadherin) and intercellular adhesion molecule-1 (ICAM1) have been reported to promote tumor infiltration of CD8 T cells and favor their anti-tumor effects[30–33]. Vascular cell adhesion molecule-1 (VCAM1) is expressed by endothelial cells, immune cells, and cancer cells[34–37]. High expression of VCAM1 is associated with tumor immune evasion in both acute myeloid leukemia (AML) and solid tumors[36,37]. In the study on AML, VCAM1 provides a 'don't eat me' signal to phagocytes, but its receptor on AML cells has not been revealed[37]. On the other hand, VCAM1 interacts with α4β1 integrin (CD49d/CD29) and promotes transendothelial migration of leukocytes into tissues[35]. These findings link the VCAM1 molecule with immune cell motility. Despite its role in enhancing cell migration, high expression of VCAM1 is associated with failure of anti-tumor responses and immune exclusion of CD8 T cells in solid tumors[38]. These studies indicate unknown function of VCAM1 in controlling tumor immunity.

Here, our study establishes the importance of motile regulation of iNKT cells in tumors. Macrophages and VCAM1 molecule in tumors respectively inhibit tumor infiltration and intratumoral motility of iNKT cells. Impaired motility interferes with antigen scanning and activation and thus leads to dysfunction of intratumoral iNKT cells. Enhancing the infiltration and motility of intratumoral iNKT cells via targeting on macrophages and VCAM1-CD49d signaling efficiently enhance efficacy of iNKT cell-based immunotherapy.

## Results

### Intratumoral iNKT cells have impaired motility and fail to scan and respond to antigen

To explore the motile regulation of iNKT cell dysfunction in tumors, we revealed their distribution and motility using intravital imaging. The distribution of iNKT cells in MC38-mCherry tumors in Vα14 Tg Cxcr6^Gfp mice was examined, and GFP⁺ iNKT cells were accumulated mainly at the boundary of tumors (Fig. 1a). With the slices of tumors, we confirmed that the densities of iNKT cells inside tumors (about 100 μm from the margin) were much lower than they were at tumor periphery (Fig. 1b). In addition to the insufficient tumor infiltration, we found that the intratumoral iNKT cells in 10 days MC38-mCherry tumors were crawling around small areas with low displacement lengths and velocities (Fig. 1c, d), whereas iNKT cells at tumor periphery exhibited long migration trajectories with stochastic direction and showed high displacement lengths and velocities (Fig. 1e–g and Supplementary Movie 1), demonstrating impaired motility of intratumoral iNKT cells. To further prove the influences of tumor microenvironment on iNKT cell motility, we transferred ex vivo GFP⁺ expanded iNKT cells into tumor-bearing mice and imaged these cells in tumors 48 h after transfer. Again, we found that just a minority of transferred cells infiltrated into 10 days MC38-mCherry tumors (Fig. 1h, i), and their motility in tumors was severely impaired in comparison with cells at tumor periphery (Fig. 1j–n and Supplementary Movie 2). Moreover, impaired motility of transferred iNKT cells were observed in 10 days B16F10-mCherry tumors as well as in ~5 weeks MC38-mCherry tumors (Fig. 1o–v, Supplementary Movie 3 and Supplementary Movie 4). These findings reveal that iNKT cells fail to efficiently infiltrate into tumors and tumor microenvironment significantly inhibits their motility.

Motility is essential for the antigen scanning and activation of iNKT cells in vivo[22,39]. Next, we investigated the antigen recognition by iNKT cells in tumors. CD1d presents lipid antigens to iNKT cells[1]. We generated CD1d conditional knockout mice to identify the antigen presenting cells activating intratumoral iNKT cells. In Lyz2^cre Cd1d^fl/fl mice, CD1d was deleted in macrophages but not in DCs, whereas, in Cd11c^cre Cd1d^fl/fl, CD1d was deleted in both macrophages and DCs (Fig. 2a). Therefore, the differences between these two mice were due to the deletion of CD1d in DCs. Given the dramatically reduced α-Galactosylceramide (αGC)-induced IFN-γ production in splenic iNKT cells and intratumoral iNKT cells from Cd11c^cre Cd1d^fl/fl mice but enhanced IFN-γ production in those cells from Lyz2^cre Cd1d^fl/fl mice (Fig. 2b, c, Supplementary Fig. 1a and Supplementary Fig. 2a, b), DCs were predominant antigen presenting cells activating and inducing Th1 anti-tumor response of iNKT cells in tumors as well as in spleens in vivo. Notably, although the intratumoral DCs (CD45.2⁺ Ly-6C⁻ CD11b⁺ MHCII⁺ F4/80⁻ CD11c⁺) in vitro presented lipid antigen αGC and activated hepatic iNKT cells as efficiently as splenic DCs (CD45.2⁺ CD11c⁺ MHCII⁺) did (Fig. 2d, Supplementary Fig. 1b, c and Supplementary Fig. 2c), as indicated by similar IFN-γ production in those hepatic iNKT cells, we found that, in vivo, iNKT cells formed clusters with DCs more efficiently in spleens than in tumors after αGC injection (Fig. 2e). Consistently, these intratumoral iNKT cells showed less IFN-γ production in comparison with splenic iNKT cells in vivo (Fig. 2f and Supplementary Fig. 2d). Given the impaired motility of intratumoral iNKT cells, these results indicate failure of antigen scanning in tumors.

### Macrophage CD1d inhibit tumor infiltration of iNKT cells, iNKT-DC interactions, but not intratumoral motility of iNKT cells

In addition to iNKT cells, we found accumulation of macrophages at the boundary of MC38-mCherry tumors (Fig. 3a). The expression of CD1d in macrophages implied direct interactions with iNKT cells. To investigate the impacts of CD1d mediated macrophage-iNKT cell interactions, we used Lyz2^cre Cd1d1^fl/fl mice. We found that CD1d deficiency in macrophages significantly increased infiltration of GFP⁺ transferred iNKT cells into MC38-mCherry tumors. With flow cytometry, we further proved that infiltration of GFP⁺ transferred iNKT cells as well as GFP⁻ host iNKT cells were increased in MC38 tumors by deleting CD1d in macrophages (Fig. 3b). In contrast to the reduced interactions between GFP⁺ transferred iNKT cells and macrophages in MC38-mCherry tumors of Lyz2^cre Cd1d1^fl/fl mice, those GFP⁺ transferred iNKT cells interacted more frequently with DCs (Fig. 3c–f), and that was in line with the increased IFN-γ production from intratumoral iNKT cells in these mice (Fig. 2c). However, deficiency of CD1d in macrophages did not alter the velocities or displacement lengths of GFP⁺ transferred iNKT in MC38-mCherry tumors (Fig. 3g–k and Supplementary Movie 5). Our data indicate that CD1d mediated interactions between macrophages and iNKT cells block iNKT cell infiltration but do not influence their intratumoral motility. The increased iNKT-DC interactions caused by deleting CD1d in macrophages are likely due to absence of competition between macrophages and DCs to interact with iNKT cells. Despite the priority of DCs in antigen presentation, macrophages might interfere with iNKT-DC interactions considering their large cell numbers. Additionally, macrophages in MC38-mCherry tumors were F4/80⁺ CD11c⁻ CD206⁺, representing M2 phenotypes (Supplementary Fig. 3). Notably, M2 macrophages favor Th2 response of iNKT cells whereas DCs favor Th1 response[40,41], and that is in line with the increased Th1 response in tumors of Lyz2^cre Cd1d1^fl/fl mice after αGC injection.

### Tumor VCAM1 confines intratumoral iNKT cell motility and impairs their antigen scanning and activation

Integrins and adhesion molecules have been previously reported to control the immune cell migration[28,29,42]. We detected higher level of Vcam1 mRNA but lower level of Icam1 mRNA in MC38 tumors than in spleens and livers (Fig. 4a). On the other hand, iNKT cells in these tumors expressed higher level of CD49d known as α4 integrin[35], a subunit of receptor for VCAM1, than those iNKT cells in spleens and livers (Fig. 4b). These findings led to a hypothesis that VCAM1 in

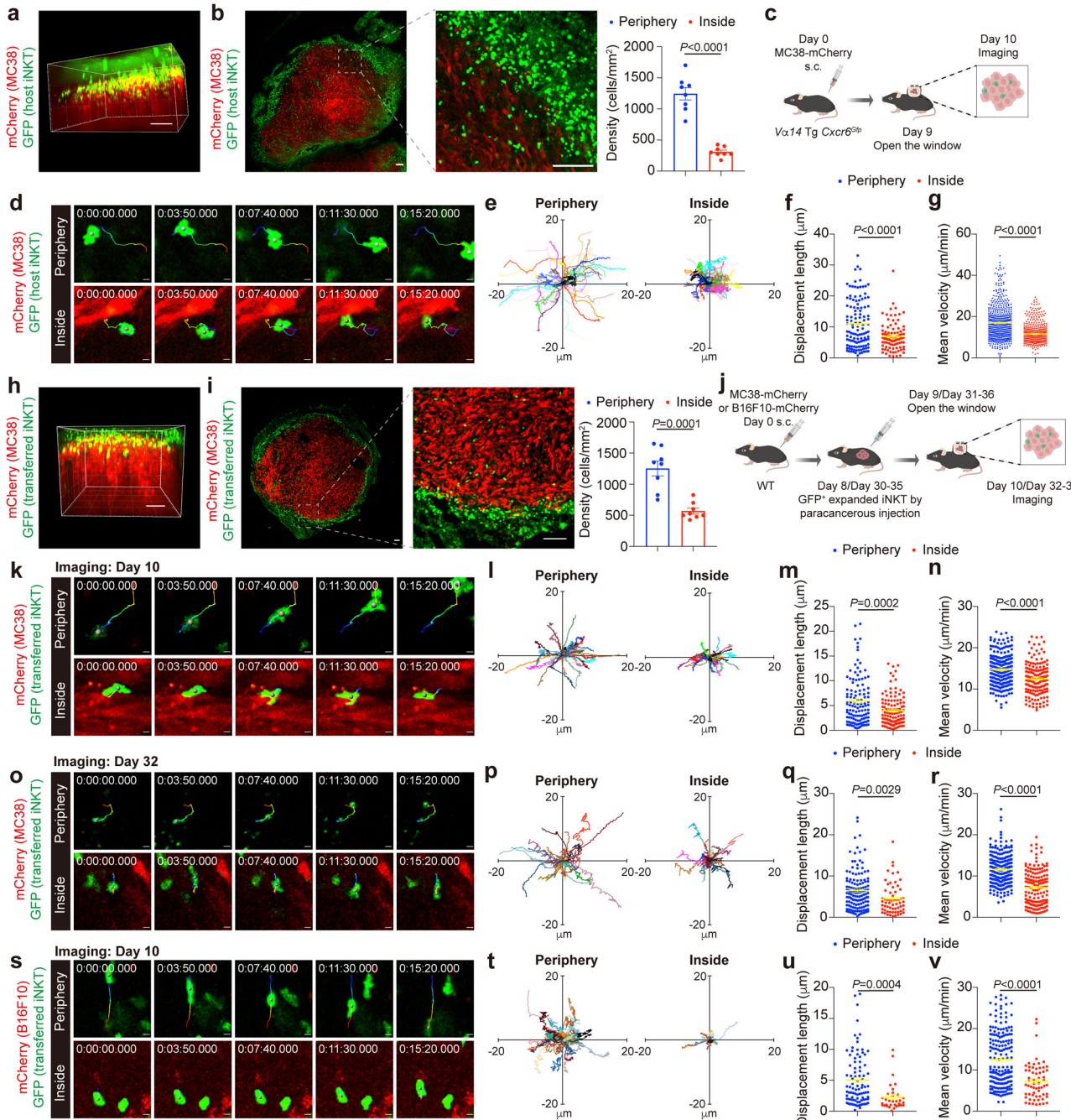

**Fig. 1 | iNKT cells in tumors exhibit low density and impaired motility.**
**a** Representative intravital 3D image of GFP⁺ host iNKT cells in MC38-mCherry tumor from *Vα14* Tg *Cxcr6^Gfp* mice. Scale Bar, 100 μm. **b** Representative images and density (*n* = 8 from 4 mice) of GFP⁺ host iNKT cells at periphery and inside tumor (about 100 μm from the margin) in MC38-mCherry tumor slices from *Vα14* Tg *Cxcr6^Gfp* mice. Scale Bars, 100 μm. **c** Schematic of the intravital imaging of GFP⁺ host iNKT cells in tumors from *Vα14* Tg *Cxcr6^Gfp* mice. **d**–**g** Time-lapse intravital images (**d**), trajectories (**e**), displacement length (**f**, *n* = 120, 93 cells from 4 mice), and mean velocities (**g**, *n* = 538, 596 cells from 4 mice) of GFP⁺ host iNKT cells at periphery or inside MC38-mCherry tumor. Scale Bars, 5 μm. **h** Representative intravital 3D image of GFP⁺ transferred iNKT cells in MC38-mCherry tumor from WT mice. Scale Bar, 100 μm. **i** Representative images and density (*n* = 8 from 4 mice) of GFP⁺ transferred iNKT cells at periphery and inside tumor (about 100 μm from the margin) in MC38-mCherry tumor slices from WT mice. Scale Bars, 100 μm. **j** Schematic of the intravital imaging of GFP⁺ transferred iNKT cells in tumors from WT mice.

**k**–**n** Time-lapse intravital images (**k**), trajectories (**l**), displacement length (**m,** *n* = 118, 118 cells from 4 mice), and mean velocities (**n**, *n* = 219, 160 cells from 4 mice) of GFP⁺ transferred iNKT cells at periphery or inside MC38 tumor of 10 days old. Scale Bars, 5 μm. **o**–**r** Time-lapse intravital images (**o**), trajectories (**p**), displacement length (**q**, *n* = 138, 62 cells from 3 mice), and mean velocities (**r**, *n* = 241, 156 cells from 3 mice) of GFP⁺ transferred iNKT cells at periphery or inside MC38-mCherry tumor of ~5 weeks old. Scale Bars, 5 μm. **s**–**v** Time-lapse intravital images (**s**), trajectories (**t**), displacement length (**u**, *n* = 92, 33 cells from 3 mice), and mean velocities (**v**, *n* = 190, 61 cells from 3 mice) of GFP⁺ transferred iNKT cells at periphery or inside B16F10-mCherry tumor of 10 days old. Scale Bars, 5 μm. Data are representative of (**a**, **b**, **d**, **h**, **i**, **k**, **o**, **s**) or pooled from (**b**, **e**–**g**, **i**, **l**–**n**, **p**–**r**, **t**–**v**) three to four independent experiments. Data are represented as mean ± SEM, and were analyzed by two-tailed unpaired Student's *t*-test (**b**, **f**, **g**, **i**, **m**–**n**, **q**–**r**, **u**-**v**). Source data are provided as a Source Data file.

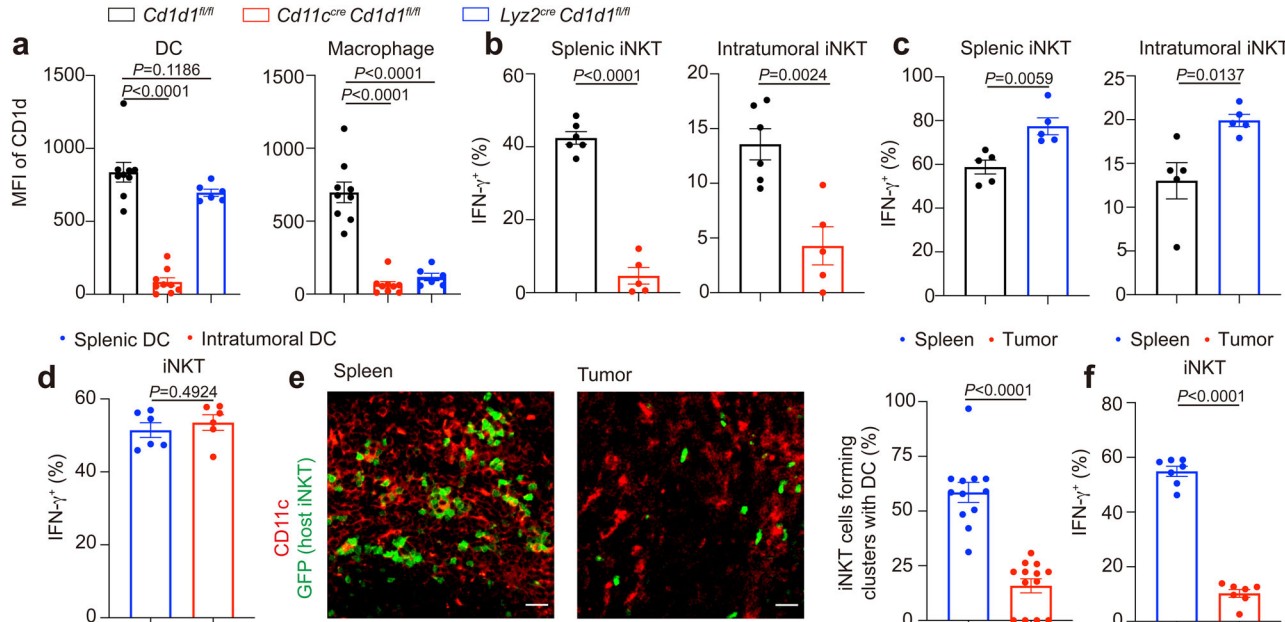

**Fig. 2 | iNKT cells fail to interact with DCs with normal antigen presentation capacity in tumors. a** CD1d expression in DCs and macrophages from *Cd1d^fl/fl* mice, *Cd11c^cre Cd1d^fl/fl* mice, and *Lyz2^cre Cd1d^fl/fl* mice (*n* = 6–9 mice per group). **b** IFN-γ production in splenic iNKT cells and intratumoral iNKT cells from MC38 tumor-bearing *Cd11c^cre Cd1d^fl/fl* mice and *Cd1d^fl/fl* mice (*n* = 5–6 mice per group), after αGC injection. αGC, α-Galactosylceramide. **c** IFN-γ production in splenic iNKT cells and intratumoral iNKT cells from MC38 tumor-bearing *Lyz2^cre Cd1d^fl/fl* mice and *Cd1d^fl/fl* mice (*n* = 5 mice per group), after αGC injection. **d** IFN-γ

production in hepatic iNKT cells stimulated with αGC-pulsed splenic DCs or αGC-pulsed intratumoral DCs for 12 h in vitro (*n* = 6 mice). **e, f** Representative images and frequencies of iNKT cells forming clusters with DC cells (**e**, *n* = 12, 13 from 4 mice), and IFN-γ production in iNKT cells (**f**, *n* = 7 mice) in spleens and tumors after αGC injection. Scale Bars, 20 μm. Data are representative of (**e**) or pooled from (**a–f**) two to four independent experiments. Data are represented as mean ± SEM or were analyzed by one-way ANOVA (**a**) and two-tailed unpaired Student's *t*-test (**b–f**). Source data are provided as a Source Data file.

tumors might influence the motility of iNKT cells. Next, we measured expression patterns of VCAM1 in tumors, and found that MC38 tumor cells expressed highest level of VCAM1, followed by monocytes, whereas DCs and particularly macrophages expressed low VCAM1 (Fig. 4c, d). We then knocked down *Vcam1* in MC38-mCherry tumor cells to evaluate its effects on iNKT cell behavior (Supplementary Fig. 4a). Notably, knockdown of *Vcam1* increased apoptosis of MC38 tumor cells (Supplementary Fig. 4b, c). To exclude the possible influence of tumor size on iNKT cell motility, we subcutaneously injected $1.5 \times 10^6$ *Vcam1* knockdown MC38-mCherry tumor cells and $1 \times 10^6$ none target control (NTC) cells in the following study to ensure similar tumor size while iNKT cell transfer (Supplementary Fig. 4d). We found that knockdown of *Vcam1* in MC38-mCherry tumor cells improved infiltration of GFP⁺ transferred iNKT cells into tumors (Fig. 4e–f). The frequencies of both GFP⁺ transferred iNKT cells and GFP⁻ host iNKT cells were elevated in *Vcam1* knockdown MC38-mCherry tumors (Fig. 4g). Additionally, we found that knockdown of *Vcam1* in MC38-mCherry tumor cells recovered motility of intratumoral GFP⁺ transferred iNKT cells, as indicated by longer migration trajectories, stochastic directions, higher velocities, and longer displacement lengths (Fig. 4h–l and Supplementary Movie 6). In line with the increased infiltration and improved motility of iNKT cells in tumors, interactions between DCs and GFP⁺ transferred iNKT cells (Fig. 4m, n) as well as the IFN-γ production in GFP⁺ transferred iNKT cells and GFP⁻ host iNKT cells (Fig. 4o) were all increased in *Vcam1* knockdown MC38-mCherry tumors. These data indicate that tumor VCAM1 confines iNKT cell motility and hinders their antigen scanning and activation in tumors.

### Tumor VCAM1 impairs intratumoral iNKT cell motility and activation via reducing CDC42 expression

To understand the mechanisms by which tumor VCAM1 controlled behavior of intratumoral iNKT cells, we sorted iNKT cells from *Vcam1*

knockdown MC38 tumors and NTC MC38 tumors and compared their transcriptomes. Knockdown of *Vcam1* in tumor cells led to different gene expression patterns in intratumoral iNKT cells, with 267 genes were upregulated and 291 genes were downregulated (Fig. 5a–c). GO-term analysis with elevated genes indicated pathways related to cell adhesion, cell migration, immunological synapse formation, and small GTPase signaling, including Rho family (Fig. 5d–e). Elevated expression of *Acvr1*, *Baiap2* and *Fgd4* in intratumoral iNKT cells from *Vcam1* knockdown MC38 tumors indicated improved CDC42 signaling, which is a Rho family small GTPase[43–45]. In line with previous study[46], we found that expression of CDC42 was reduced in intratumoral iNKT cells (Fig. 5f). Notably, knockdown of tumor *Vcam1* increased CDC42 expression in both GFP⁺ transferred and GFP⁻ host iNKT cells in tumors (Fig. 5g). CD49d is a subunit of receptor for VCAM1. We showed that MC38 tumor cells downregulated CDC42 expression in iNKT cells in vitro, and that was restored by antibodies against either VCAM1 or CD49d (Fig. 5h). These data confirm that tumor VCAM1 inhibits CDC42 expression in iNKT cells through CD49d. Additionally, we found that MC38 tumor cells reduced phosphorylation of Scr kinase in iNKT cells in vitro, and anti-CD49d antibody partially restored the Src phosphorylation (Supplementary Fig. 5a). Via inhibiting Src activity with inhibitor KX-01, we proved that reduced Src activity led to down regulation of CDC42 expression in iNKT cells (Supplementary Fig. 5b). Our data indicate that Src signal pathway is involved in the VCAM1-CD49d signaling induced down regulation of CDC42. On the other hand, we excluded the influence of CD1d-mediated macrophage-iNKT cell interactions on CDC42 expression in GFP⁺ transferred and GFP⁻ host iNKT cells in tumors (Fig. 5i).

CDC42 is known to control the cell migration and immunological synapse formation[46,47]. With CDC42 inhibitor ZCL278, we proved that CDC42 activity was required for normal iNKT cell motility such as stochastic migration, high velocities, and long displacement lengths in vitro (Fig. 6a–d). Furthermore, we showed that interactions between

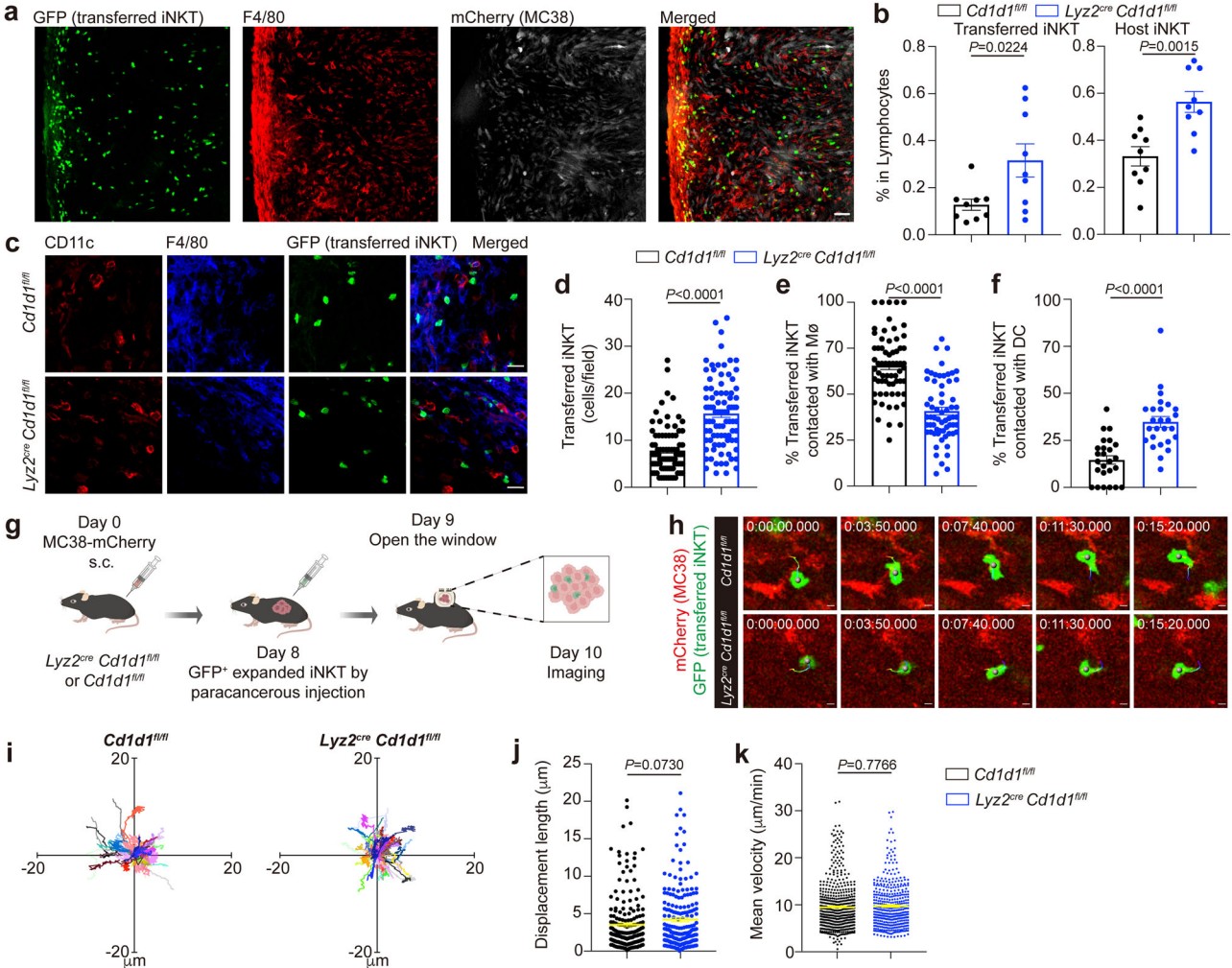

**Fig. 3 | Macrophage CD1d inhibits tumor infiltration of iNKT cells and iNKT-DC interactions but not the intratumoral motility of iNKT cells. a** Representative images showing enrichment of GFP⁺ transferred iNKT cells and F4/80⁺ macrophages at tumor boundary from WT mice. Scale Bar, 50 μm. **b** Frequencies of GFP⁺ transferred and GFP⁻ host iNKT cells in MC38-mCherry tumors from *Lyz2^cre Cd1d^fl/fl* mice and *Cd1d^fl/fl* mice detected by flow cytometry ($n = 9$ mice per group). **c–f** Representative images (**c**), numbers of GFP⁺ transferred iNKT cells (**d**, $n = 91, 88$ from 5 mice per group), frequencies of GFP⁺ transferred iNKT cells contacting with F4/80⁺ macrophages (**e**, $n = 65, 63$ from 5 mice per group), and frequencies of GFP⁺ transferred iNKT cells contacting with DC (**f**, $n = 25, 26$ from 5 mice per group) in fields of MC38-mCherry tumor slices from *Lyz2^cre Cd1d^fl/fl* mice and *Cd1d^fl/fl* mice after αGC injection. Scale Bars, 20 μm. **g**, Schematic of intravital imaging of GFP⁺ transferred iNKT cells in MC38-mCherry tumors in *Lyz2^cre Cd1d^fl/fl* mice and *Cd1d^fl/fl* mice. **h–k** Time-lapse intravital images (**h**), trajectories (**i**), displacement length (**j**, $n = 204, 199$ cells from 4 mice per group), and mean velocities (**k**, $n = 541, 436$ cells from 4 mice per group) of GFP⁺ transferred iNKT cells in MC38-mCherry tumors in *Lyz2^cre Cd1d^fl/fl* mice and *Cd1d^fl/fl* mice. Scale Bars, 5 μm. Data are represented as mean ± SEM. Data are representative of (**a, c, h**) or pooled from (**b, d–f, i–k**) three to six independent experiments. Data were analyzed by two-tailed unpaired Student's *t*-test (**b, d, e, f, j, k**). Source data are provided as a Source Data file.

αGC-pulsed BMDCs and iNKT cells, their synapse lengths, as well as IFN-γ production in iNKT cells were all CDC42 activity dependent (Fig. 6e–h). To further prove that CDC42 reduction in iNKT cells led to impaired motility and anti-tumor function, we overexpressed CDC42 in iNKT cells. We found that MC38 tumor cells inhibited motility of iNKT cells in vitro via a VCAM1 dependent manner (Fig. 6i–k), and that was restored by overexpression of CDC42 in iNKT cells (Fig. 6l–n). Additionally, we confirmed that, in vivo, CDC42 overexpression led to enhanced tumor infiltration, IFN-γ production, and anti-tumor efficacy of iNKT cells (Fig. 6o–s). Together, our data indicate that tumor VCAM1 inhibits iNKT cell motility and activation via reducing CDC42 expression.

### Human tumor VCAM1 impairs human iNKT cell motility and activation via reducing CDC42 expression

With data in TCGA, we found that low grade gliomas (LGG) patients with higher tumor VCAM1 expression exhibited lower survival rates (Fig. 7a), and VCAM1 level was negatively correlated with level of iNKT cell signature genes, *ZBTB16* and *TRAV24*, indicating poor iNKT cell infiltration in VCAM1 high expression LGG patients (Fig. 7b). Human iNKT cells expressed CD49d as well (Fig. 7c), implying possible roles of VCAM1-CD49d signaling in controlling their motility and activation. Indeed, we showed that VCAM1 overexpressing MDA-MB-231.hVCAM1 cells reduced stochastic migration, velocities, and displacement lengths of human iNKT cells in vitro, and antibody against VCAM1 restored the motility of human iNKT cells (Fig. 7d–h). In agreement with the role of VCAM1-CD49d signaling in controlling CDC42 expression in mouse iNKT cells, we showed that human MDA-MB-231.hVCAM1 cells reduced expression of CDC42 in human iNKT cells in a VCAM1 dependent manner (Fig. 7i). Again, inhibiting CDC42 activity in human iNKT cells by ZCL278 impaired their motility in vitro (Fig. 7j–m). When human iNKT cells were activated by artificial antigen presenting cells HeLa.hCD1d in the presence of αGC, ZCL278 decreased iNKT-HeLa.hCD1d interactions and synapse lengths

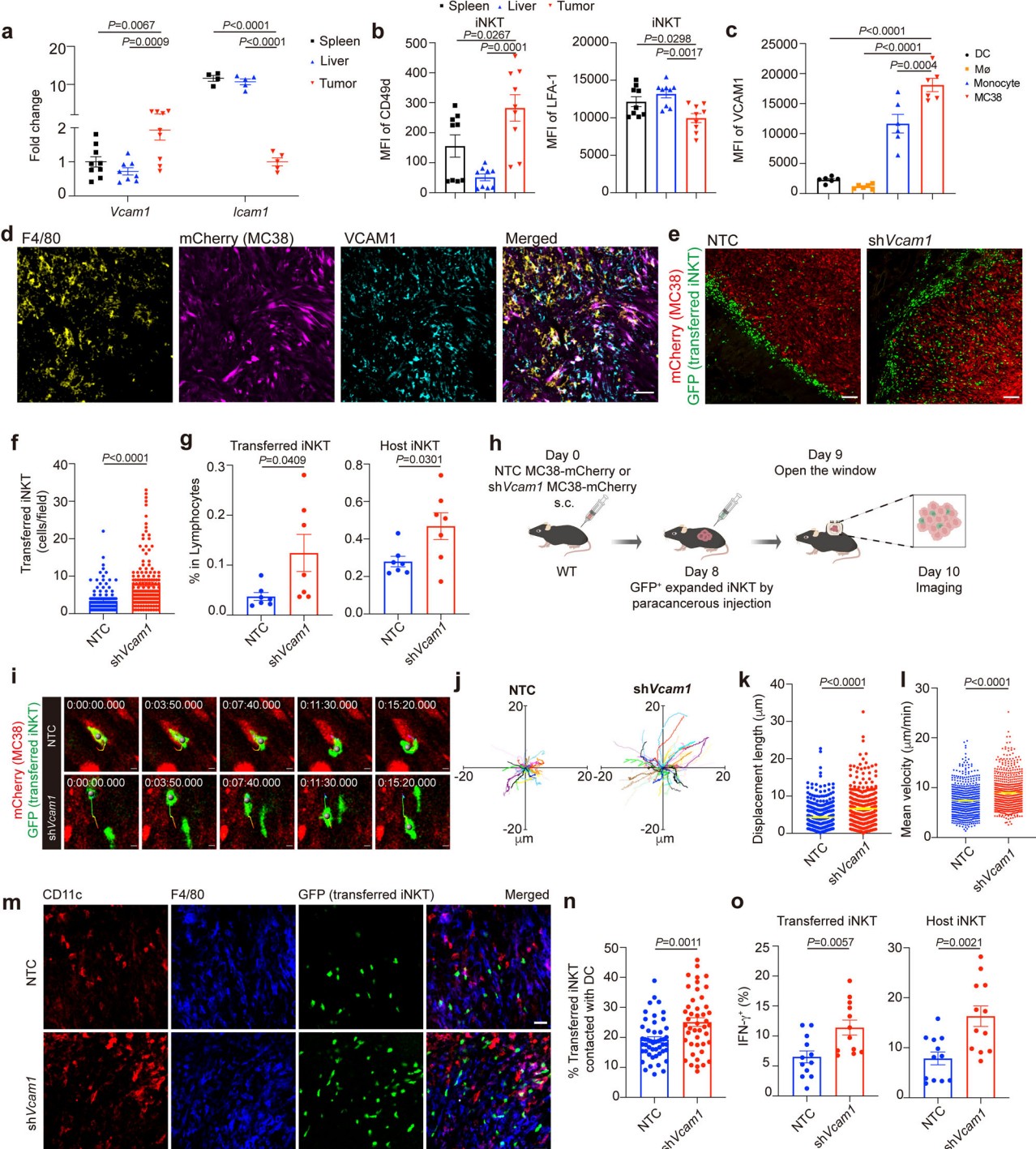

(Fig. 7n–p), and reduced αGC-induced IFN-γ production (Fig. 7q). Next, we tested whether VCAM1 in human tumor cells inhibited human iNKT cell activation (Fig. 7r). We found that human MDA-MB-231.hVCAM1- cells interfered with interactions between human iNKT cells and αGC- presenting HeLa.hCD1d cells, artificial antigen presenting cells with high CD1d expression (Fig. 7r–t), reduced their synapse lengths, inhibited the IFN-γ production (Fig. 7u–w), and all these were restored by anti-VCAM1 antibody (Fig. 7r–w). MDA-MB-231.hVCAM1 cells expressed no CD1d (Fig. 7s). Given the fact that some tumor cells express CD1d molecule, we next generated CD1d expressing MDA-MB-231.hVCAM1.hCD1d cells (Supplementary Fig. 6a). Notably, in vivo, DCs predominantly present antigens to iNKT cells, despite expression of CD1d by other immune and non-immune cells, and that is due to the

high antigen presenting capacity of DCs[5]. Even tumor cells express CD1d molecule, they could not present lipid antigens as efficiently as DCs do. For this reason, we used MDA-MB-231.hVCAM1.hCD1d tumor cells with median CD1d expression level, and used CD1d high expressing Hela.hCD1d cells to mimic DCs with high antigen pre- senting capacity. Again, we found that MDA-MB-231.hVCAM1.hCD1d cells inhibited interactions between human iNKT cells and αGC- presenting HeLa.hCD1d cells (Supplementary Fig. 6b–e), reduced their synapse lengths, diminished the IFN-γ production, and did so in a VCAM1 dependent manner (Supplementary Fig. 6b–e). These data demonstrate similar role of tumor VCAM1 in inhibiting human iNKT cell motility and activation irrespective of CD1d expression in tumor cells.

**Fig. 4 | VCAM1 in tumor cells inhibits tumor infiltration, intratumoral motility, and antigen scanning of iNKT cell. a** The mRNA levels of *Vcam1* (n = 9 mice) and *Icam1* (n = 5 mice) in spleens, livers, and MC38 tumors from WT mice. **b** The expression of CD49d and LFA-1 in splenic iNKT cells, hepatic iNKT cells, and intratumoral iNKT cells from WT mice (n = 9 mice). **c** The VCAM1 expression in DCs, macrophages (Mø), monocytes, and tumor cells in MC38 tumors from WT mice (n = 6 mice). **d** Representative images showing VCAM1 expression in F4/80⁺ macrophages and MC38-mCherry tumor cells. Scale Bar, 50 μm. **e, f** Representative images (**e**) and numbers (**f**, n = 138, 156 from 5 mice per group) of GFP⁺ transferred iNKT cells in each field of *Vcam1* knockdown MC38-mCherry tumor slices or NTC MC38-mcherry tumor slices from WT mice. Scale Bars, 100 μm. NTC, none target control cells, transfected with scramble shRNA. **g** Frequencies of intratumoral GFP⁺ transferred and GFP⁻ host iNKT cells in *Vcam1* knockdown MC38 tumors and NTC MC38 tumors from WT mice detected by flow cytometry (n = 7 mice per group). **h** Schematic of intravital imaging of GFP⁺ transferred iNKT cells in *Vcam1*

knockdown MC38-mCherry tumors and NTC MC38-mCherry tumors from WT mice. **i–l** Time-lapse intravital images (**i**), trajectories (**j**), displacement length (**k**, n = 331, 317 cells from 5 mice per group), and mean velocities (**l**, n = 578, 633 cells from 5 mice per group) of GFP⁺ transferred iNKT cells in *Vcam1* knockdown MC38-mcherry tumors and NTC MC38-mCherry tumors. Scale Bars, 5 μm. **m, n** Representative images (**m**) and frequencies (**n**, n = 48, 45 from 5 mice per group) of GFP⁺ transferred iNKT cells contacted with DCs (F4/80⁻ CD11c⁺) in fields of *Vcam1* knockdown MC38-mCherry tumor slices and NTC MC38-mCherry tumor slices from WT mice after αGC injection. Scale Bars, 30 μm. **o** IFN-γ production in intratumoral GFP⁺ transferred and GFP⁻ host iNKT cells in *Vcam1* knockdown MC38-mCherry tumors and NTC MC38-mCherry tumors from WT mice after αGC injection (n = 12 mice per group). Data are representative of (**d, e, i, m**) or pooled from (**a, b, f, g, j–l, n, o**) two to five independent experiments. Data are represented as mean ± SEM, and were analyzed by one-way ANOVA (**a–c**) and unpaired Student's *t*-test (**f, g, k, l, n, o**). Source data are provided as a Source Data file.

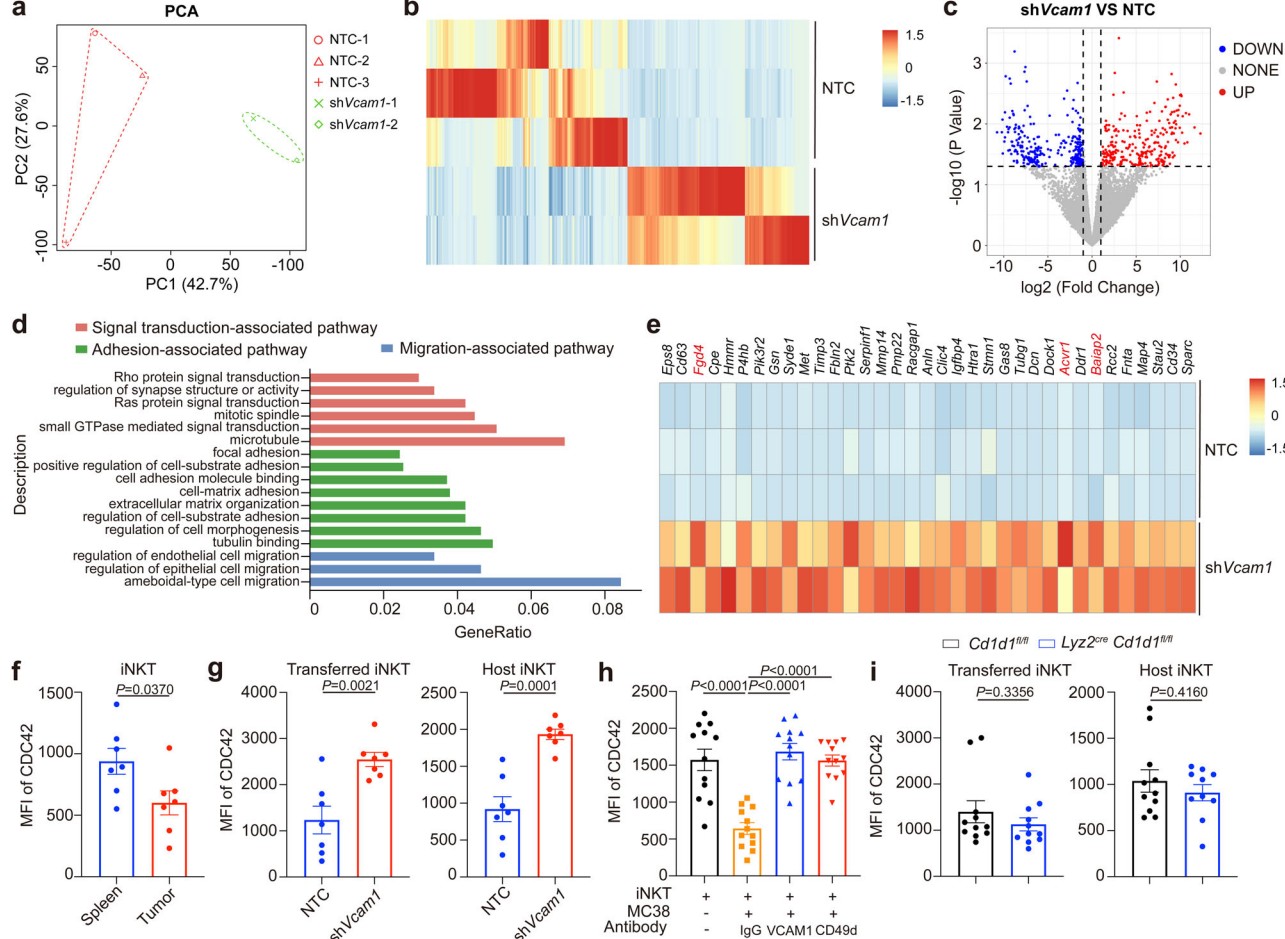

**Fig. 5 | Tumor VCAM1 reduced CDC42 expression in intratumoral iNKT cells.** **a** Principal component analysis (PCA) plots for RNA-seq data of intratumoral iNKT cells sorted from *Vcam1* knockdown MC38 tumor-bearing WT mice and NTC MC38 tumor-bearing WT mice (n = 2–3 mice per group). **b** Heatmap showing genes distinctly expressed in iNKT cells sorted from *Vcam1* knockdown MC38 tumors and NTC MC38 tumors (Fold Change ≥ 2 or ≤ 0.5, P-value < 0.05). **c** Volcano plot showing genes up-regulated and down-regulated in iNKT cells from **b**. **d** GO-term analysis of genes up-regulated in iNKT cells sorted from *Vcam1* knockdown MC38 tumors in comparison with iNKT cells from NTC MC38 tumors. **e** Heatmap showing genes in (**d**). **f** CDC42 expression in splenic iNKT cells and intratumoral iNKT cells from MC38 tumor-bearing WT mice (n = 7 mice). **g** CDC42 expression in GFP⁺

transferred and GFP⁻ host iNKT cells in *Vcam1* knockdown MC38 tumors and NTC MC38 tumors from WT mice (n = 7 mice per group). NTC, none target control. **h** CDC42 expression in expanded iNKT cells co-cultured with or without MC38 tumor cells in the presence of anti-VCAM1 antibody, anti-CD49d antibody, or IgG isotype control antibody for 24 h (n = 12 per group). **i** CDC42 expression in GFP⁺ transferred and GFP⁻ host iNKT cells in MC38 tumors from *Lyz2^cre^ Cd1d1^fl/fl^* mice and *Cd1d1^fl/fl^* mice (n = 10–11 mice per group). Data are pooled from three to five independent experiments (**f–i**). Data are represented as mean ± SEM, and were analyzed by Fisher's exact test (**b-c**), two-tailed unpaired Student's *t*-test (**f, g, i**) and one-way ANOVA (**h**). Source data are provided as a Source Data file.

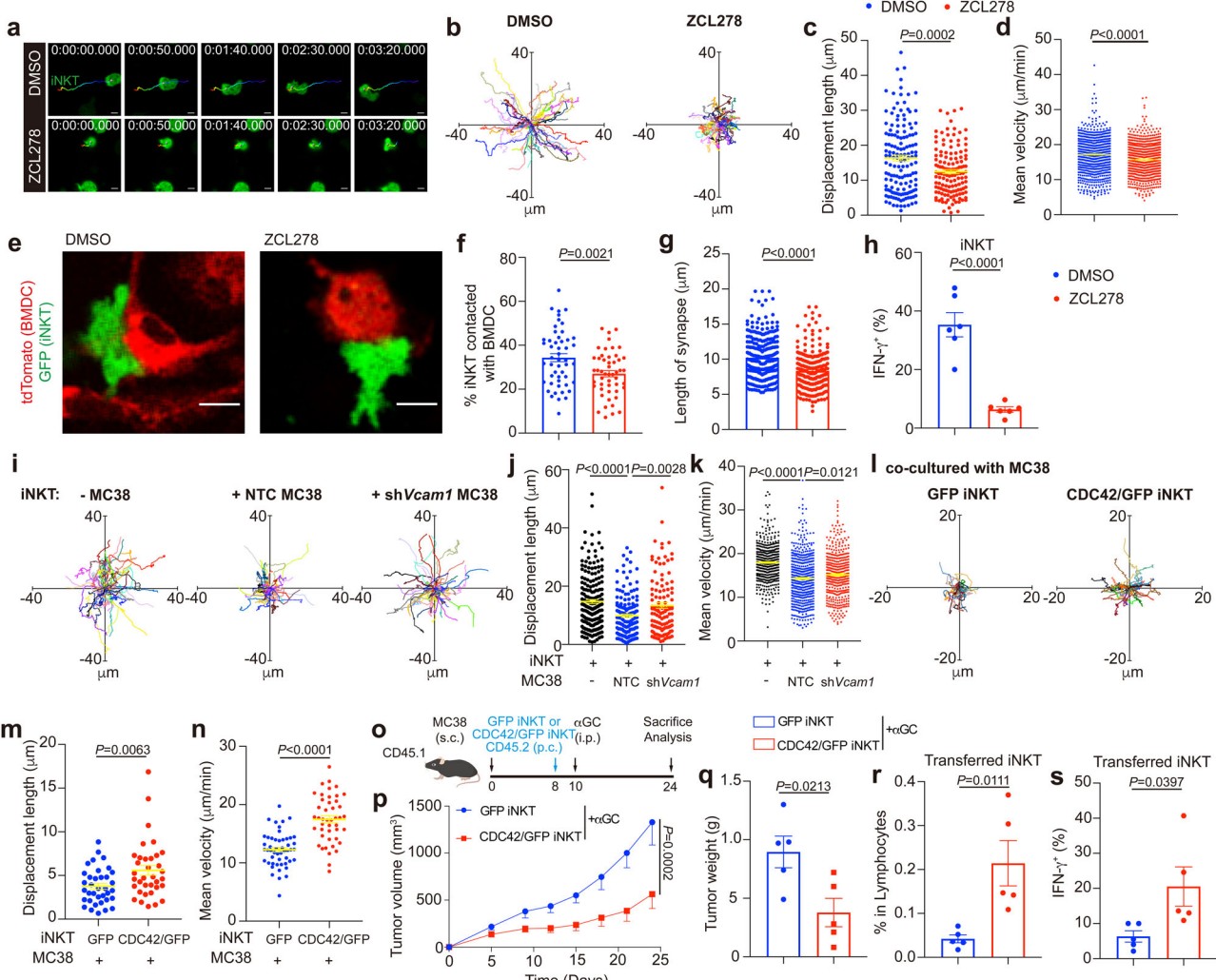

**Fig. 6 | Reduced CDC42 expression induced by tumor VCAM1 inhibits motility and activation of intratumoral iNKT cells. a–d** Time-lapse images (**a**), trajectories (**b**), displacement lengths (**c**, $n = 146, 134$ cells), and mean velocities (**d**, $n = 923, 998$ cells) of DMSO or ZCL278 pretreated GFP⁺ expanded iNKT cells in vitro. Scale Bars, 5 µm. **e–g** Images (**e**), frequencies (**f**, $n = 50, 50$), and lengths of synapses (**g**, $n = 268$, 221 cells) of GFP⁺ expanded iNKT cells contacted with αGC-pulsed tdTomato⁺ BMDCs in the presence of DMSO or ZCL278. Scale Bars, 10 µm. **h** IFN-γ production in GFP⁺ expanded iNKT cells stimulated with αGC-pulsed BMDCs in the presence of DMSO or ZCL278 ($n = 6$). **i–k** Trajectories (**i**), displacement lengths (**j**, $n = 195, 156$, 113 cells), and mean velocities (**k**, $n = 479, 554, 418$ cells) of GFP⁺ iNKT cells alone or co-cultured with NTC MC38 tumor cells or sh*Vcam1* MC38 tumor cells in vitro. NTC,

none target control. **l–n** Trajectories (**l**), displacement lengths (**m**, $n = 36, 35$ cells), and mean velocities (**n**, $n = 50, 45$ cells) of iNKT cells overexpressed GFP control or CDC42/GFP in presence of MC38 tumor cells in vitro. **o** Timeline of experimental procedure for (**p–s**). **p, q** Growth curves (**p**), and weight (**q**) of MC38 tumors in WT mice receiving αGC and iNKT cells overexpressed GFP control or CDC42/GFP ($n = 5$ mice per group). **r, s** Frequencies (**r**) and IFN-γ production (**s**) of transferred iNKT cells overexpressed GFP control or CDC42/GFP in MC38 tumors described in (**o–q**). Data are representative of (**a**, **e**) or pooled from (**b–d**, **f–n**) two to five independent experiments. Data are represented as mean ± SEM, and were analyzed by two-tailed unpaired Student's *t*-test (**f–h**, **m**, **n**, **q–s**), one-way ANOVA (**j**, **k**) and two-way ANOVA (**p**). Source data are provided as a Source Data file.

## Tumor VCAM1 reduces CDC42 expression and cytokine production in mouse and human CD8 T cells

CD8 T cells are the other key players in anti-tumor immune responses[48,49]. With MC38 tumor-bearing mouse models, we found that intratumoral CD8 T cells expressed higher level of CD49d than those splenic and hepatic CD8 T cells (Supplementary Fig. 7a). When mouse CD8 T cells were co-cultured with MC38 tumor cells in vitro, they reduced CDC42 expression, and antibodies against VCAM1 or CD49d partially restored their CDC42 expression (Supplementary Fig. 7b). With ZCL278, we proved that anti-CD3 plus anti-CD28-induced IFN-γ production in these mouse CD8 T cells required CDC42 activity (Supplementary Fig. 7c, d). Moreover, we confirmed that tumor VCAM1 reduced CDC42 expression in human CD8 T cells as well (Supplementary Fig. 7e, f), and that led to impaired IFN-γ production (Supplementary Fig. 7g, h). These results demonstrate similar effects of VCAM1-CD49d signaling on CD8 T cells as on iNKT cells.

## Interfering with VCAM1-CD49d signaling enhances anti-tumor efficacy of iNKT cells

To evaluate the efficacy of iNKT cell-based therapy, we transferred expanded iNKT cells into tumor-bearing mice and injected them with αGC. iNKT cell transfer plus αGC injection significantly inhibited MC38 tumor growth (Fig. 8a–d). In line with our results that knockdown of *Vcam1* in MC38-mCherry tumor cells enhanced iNKT cell infiltration into tumors (Fig. 4g) and augmented their IFN-γ production (Fig. 4o), knockdown of tumor *Vcam1* further enhanced anti-tumor efficacy of iNKT cells (Fig. 8a–d). Notably, knockdown of *Vcam1* alone, without iNKT cell transfer plus αGC injection, impaired MC38 tumor growth, possibly due to the effect of VCAM1 on tumor cell apoptosis (Supplementary Fig. 4b, c). Since VCAM1 confined iNKT cell motility and activation through its receptor (Fig. 5h), we next used anti-CD49d antibody to interfere with VCAM1-receptor signaling together with iNKT cell transfer plus αGC injection (Fig. 8e). Anti-CD49d antibody

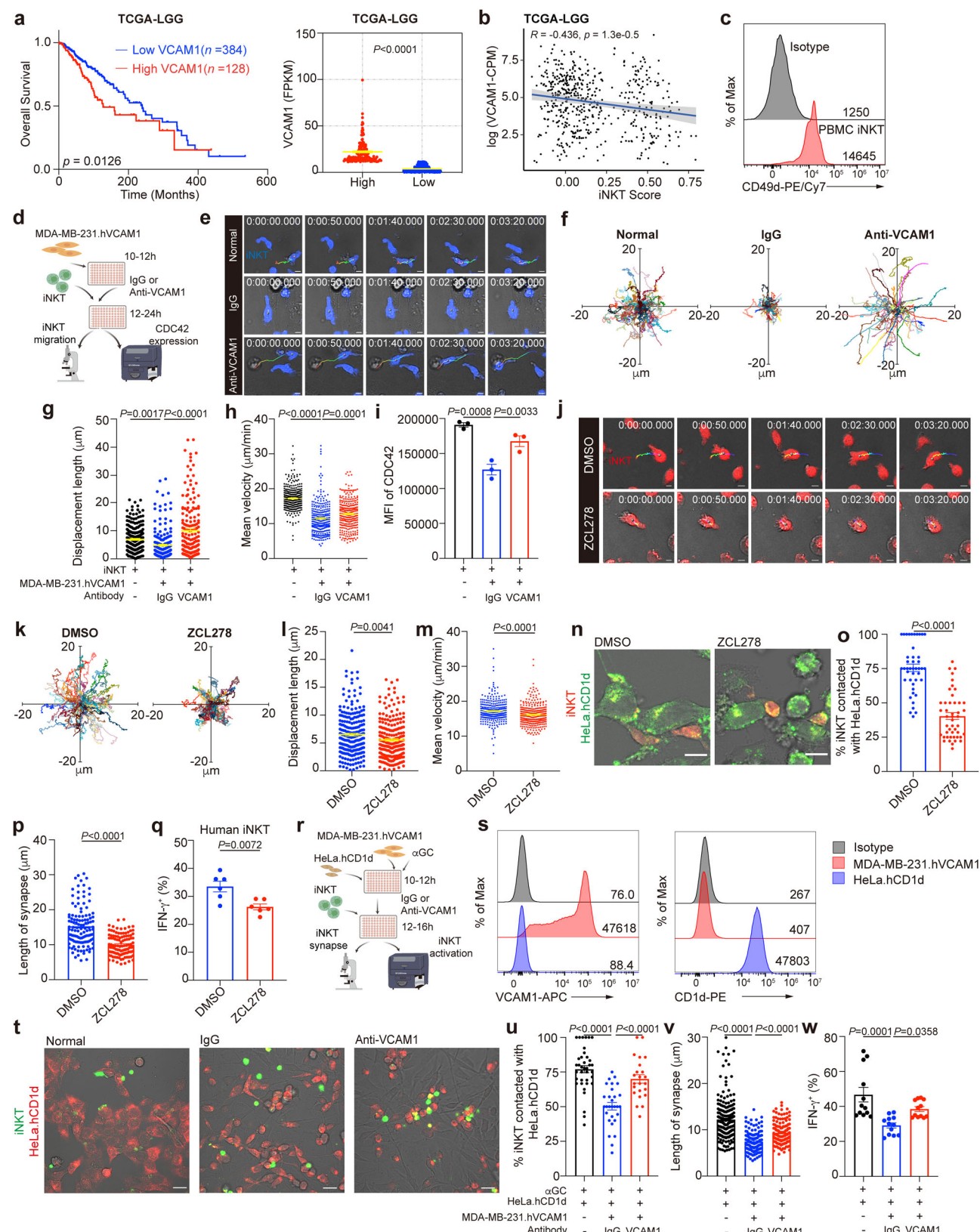

alone showed no influence on MC38 tumor growth in vivo (Fig. 8h). Both anti-CD49d antibody and anti-VCAM1 antibody did not influence MC38 tumor cell apoptosis in vitro (Supplementary Fig. 4f). These results suggest that blocking VCAM1-CD49d signaling via antibody alone would not cause tumor cell apoptosis directly. Notably, anti-CD49d antibody in combination with iNKT cell transfer plus αGC

injection significantly enhanced anti-tumor efficacy of iNKT cells, as indicated by reduced MC38 tumor growth (Fig. 8f–h), increased tumor infiltration of iNKT cells, and enhanced IFN-γ production in intratumoral iNKT cells (Fig. 8i–j), in comparison with iNKT cell transfer plus αGC injection alone. On the other hand, in addition to the influence of VCAM1-CD49d signaling on CD8 T cells (Supplementary Fig. 7b),

**Fig. 7 | Tumor cell VCAM1 inhibits human iNKT cell motility and activation.**
**a** Survival of low-grade gliomas patients with low *VCAM1* and high *VCAM1* (*n* = 384, 128 samples). Data from TCGA-LGG database. **b** Correlation between iNKT cell infiltration and *VCAM1* level in patients described in (**a**). Error bands represent 95% confidence intervals. **c** CD49d expression in human iNKT cells from PBMCs. **d** Experimental procedure for (**e–i**). **e-i**, Time-lapse images (**e**), trajectories (**f**), displacement lengths (**g**, *n* = 264, 186, 180 cells), mean velocities (**h**, *n* = 349, 290, 303 cells), and CDC42 expression (**i**, *n* = 3 samples) of CTV-stained expanded human iNKT cells co-cultured with or without MDA-MB-231.hVCAM1 cells in the presence of indicated antibodies. Scale Bars, 5 µm. **j–m** Time-lapse images (**j**), trajectories (**k**), displacement lengths (**l**, *n* = 243, 230 cells), and mean velocities (**m**, *n* = 325, 315 cells) of CMTPX-stained expanded human iNKT cells in the presence of DMSO or ZCL278. Scale Bars, 5 µm. **n–p** Images (**n**), frequencies (**o**, *n* = 43, 46), and synapse lengths (**p**, *n* = 121, 110 cells) of CMTPX-stained expanded human iNKT cells contacted with CTV-stained αGC-pulsed HeLa.hCD1d cells in the presence of DMSO or ZCL278. Scale Bars, 10 µm. **q** IFN-γ production in expanded human iNKT cells activated as in **n–p** (*n* = 6 samples). **r** Experimental procedure for (**s–w**). **s** Expression of VCAM1 and CD1d in HeLa.hCD1d cells and MDA-MB-231.hVCAM1 cells. **t–v** Images (**t**), frequencies (**u**, *n* = 37, 29, 23), and lengths of synapses (**v**, *n* = 204, 129, 132 cells) of CTV-stained expanded human iNKT cells contacted with CMTPX-stained αGC-pulsed HeLa.hCD1d cells in the absence or presence of MDA-MB-231.hVCAM1 cells and indicated antibodies. Scale Bars, 20 µm. **w** IFN-γ production in expanded human iNKT cells activated as in **t–v** (*n* = 12 samples). Data are representative of (**c, e, j, n, t**) or pooled from (**f–h, k–m, o–q, u–w**) two to three independent experiments. Data are represented as mean ± SEM, and were analyzed by log-rank test (**a**), Pearson's correlation (**b**), one-way ANOVA (**g–i, u–w**), and two-tailed unpaired Student's *t*-test (**l, m, o–q**). Source data are provided as a Source Data file.

iNKT cells have been shown to promote activation of CD8 T cell and NK cell indirectly[2,3]. Here, we showed that anti-CD49d antibody treatment also augmented tumor infiltration of CD8 T cells and NK cells (Fig. 8i) as well as their IFN-γ production (Fig. 8j).

Notably, B16F10 tumor cells were different from MC38 tumor cells and expressed low/no level of VCAM1 (Supplementary Fig. 4e), indicating discrepant VCAM1 expression in tumor cells. Similar as in MC38 tumors, we found that immune cells in B16F10 tumors expressed VCAM1 as well (Fig. 8k) and thus could mediate VCAM1-CD49d signaling in iNKT cells. This finding also explained impaired motility of intratumoral iNKT cells in B16F10-mCherry tumors. Next, we investigated whether blocking the VCAM1-CD49d signaling would enhance anti-tumor efficacy of iNKT cells against the VCAM1 low/no tumor cells (Fig. 8l–q). We found that anti-VCAM1 antibody treatment and anti-CD49d antibody treatment significantly enhanced anti-tumor efficacy of iNKT cells in B16F10 tumor models (Fig. 8m–o). Again, these antibodies showed no influence on B16F10 tumor cell apoptosis in vitro (Supplementary Fig. 4g). Additionally, we confirmed that interfering with the VCAM1-CD49d signaling in these B16F10 tumor models augmented IFN-γ production in intratumoral iNKT cells, despite no influence on cell infiltration (Fig. 8p–q). Consistently, these antibody treatments increased IFN-γ production in intratumoral CD8 T cells and NK cells, although showed no influence on their tumor infiltration (Fig. 8p–q). These results suggest that VCAM1 molecule in tumors, not limited in tumor cells, inhibits anti-tumor responses of iNKT cells.

Both MC38 and B16F10 tumor cells in our study expressed no CD1d[50,51]. However, tumor cells are known to differ in CD1d expression[52–54]. Considering the direct killing function of iNKT cells mediated by CD1d[5,6], we explored the effect of blocking VCAM1-CD49d signaling on clearance of CD1d- and VCAM1-expressing tumor cells. We generated CD1d overexpressing MC38 tumor cells. We found that CD1d overexpression in MC38 tumor cells showed no influence on anti-tumor efficacy of iNKT cells, and knockdown of *Vcam1* efficiently elevated anti-tumor efficacy of iNKT cells in CD1d expressing MC38 tumor models (Fig. 8r–u). Together, our data demonstrate that interfering with VCAM1-CD49d signaling enhances anti-tumor efficacy of iNKT cells irrespective the expression of CD1d or VCAM1 by tumor cells.

## Interfering with macrophage-iNKT cell interactions synergizes with inhibiting VCAM1-CD49d signaling to improve anti-tumor efficacy of iNKT cells

In agreement with the inhibitory effect of macrophage-iNKT cell interactions on tumor infiltration and function of iNKT cells, interfering with these cell interactions increased anti-tumor efficacy of iNKT cells (Fig. 9a–d). Notably, interfering with macrophage-iNKT cell interactions in addition to knocking down *Vcam1* in MC38 tumor cells further enhanced anti-tumor efficacy of iNKT cells (Fig. 9a–d). Next, we investigated whether depleting macrophages could synergize with anti-VCAM1 antibody to augment anti-tumor efficacy of iNKT cells.

PLX3397, an inhibitor of colony-stimulating factor-1 receptor (CSF-1R)[55], was used to delete macrophages. Indeed, PLX3397 and anti-VCAM1 in combination better enhanced anti-tumor efficacy of transferred expanded iNKT cells than PLX3397 alone and anti-VCAM1 alone (Fig. 9e–g). These results indicate a way to improve efficacy of iNKT cell-based immunotherapy via blocking VCAM1-CD49d signaling and targeting on macrophage in combination.

## Discussion

Given the advantages of iNKT cells in anti-tumor immunotherapies, dysfunction of these cells in tumors has drawn a lot of attention. Lactic acid, impaired lipid synthesis, and limited glycolysis could all contribute to the impaired function of intratumoral iNKT cells[7,8,16]. However, this metabolic regulation focusing on dysfunctional status and impaired intracellular signal cascades represents a static regulation paradigm. Different from other tissue cells, immune cells are migrating in tissues for antigen searching and the immune surveillance[20,29,39,40]. Motility has been considered as an essential element for immune cell activation and proper immune responses[21,22,24,25,56]. Here, we report impaired motility of intratumoral iNKT cells. In tumors, DCs are major antigen presenting cells for activating iNKT cells and particularly for inducing Th1 anti-tumor immune responses[2,5]. Due to the low frequency of DCs in tumors[57], confined motility of intratumoral iNKT cells hinders their antigen scanning and activation, demonstrating a motile regulation paradigm. Additionally, previous study on CD8 T cells has reported that macrophages inhibit motility of CD8 T cells in tumors and impede their anti-tumor effect[55]. Given current studies on immune exhaustion, this impaired motility might represent a motile exhaustion that leads to hyporesponsiveness[58]. Meanwhile, a recent study on tumor slices indicates faster movement of intratumoral CD8 T cells in regions with low tumor cell density and showing exhausted gene profile[59]. Our study sheds light on the influence of tumor cell density on immune cell motility. However, the relationships between tumor cell density, immune cell motility, and properties of static exhaustion, including high expression of inhibitory receptors, low expression of effect molecules, low proliferation, metabolic reprogramming, and epigenetic remodeling[13,14,58], remain unclear.

In addition to high capacity in antigen presentation, DCs are known to favor Th1 response of iNKT cells, which is important for anti-tumor immunity[2,5]. Therefore, CD1d expression in DCs is essential for iNKT cell-mediated anti-tumor immune responses. Conversely, CD1d expression in macrophages inhibits tumor infiltration of iNKT cells and interferes with iNKT-DC interactions as well as Th1 response in tumors. In vivo, CD1d expressing antigen presenting cells could compete to interact with iNKT cells, despite the priority of DCs. Published studies have demonstrated that different antigen presenting cells interact with iNKT cells and lead to distinct immune responses[41]. Notably, M2 macrophages are known to shape iNKT cell mediated responses toward Th2[5,40,41]. Given the M2 phenotype of tumor macrophages, we could not exclude the possibility that macrophage-iNKT cell

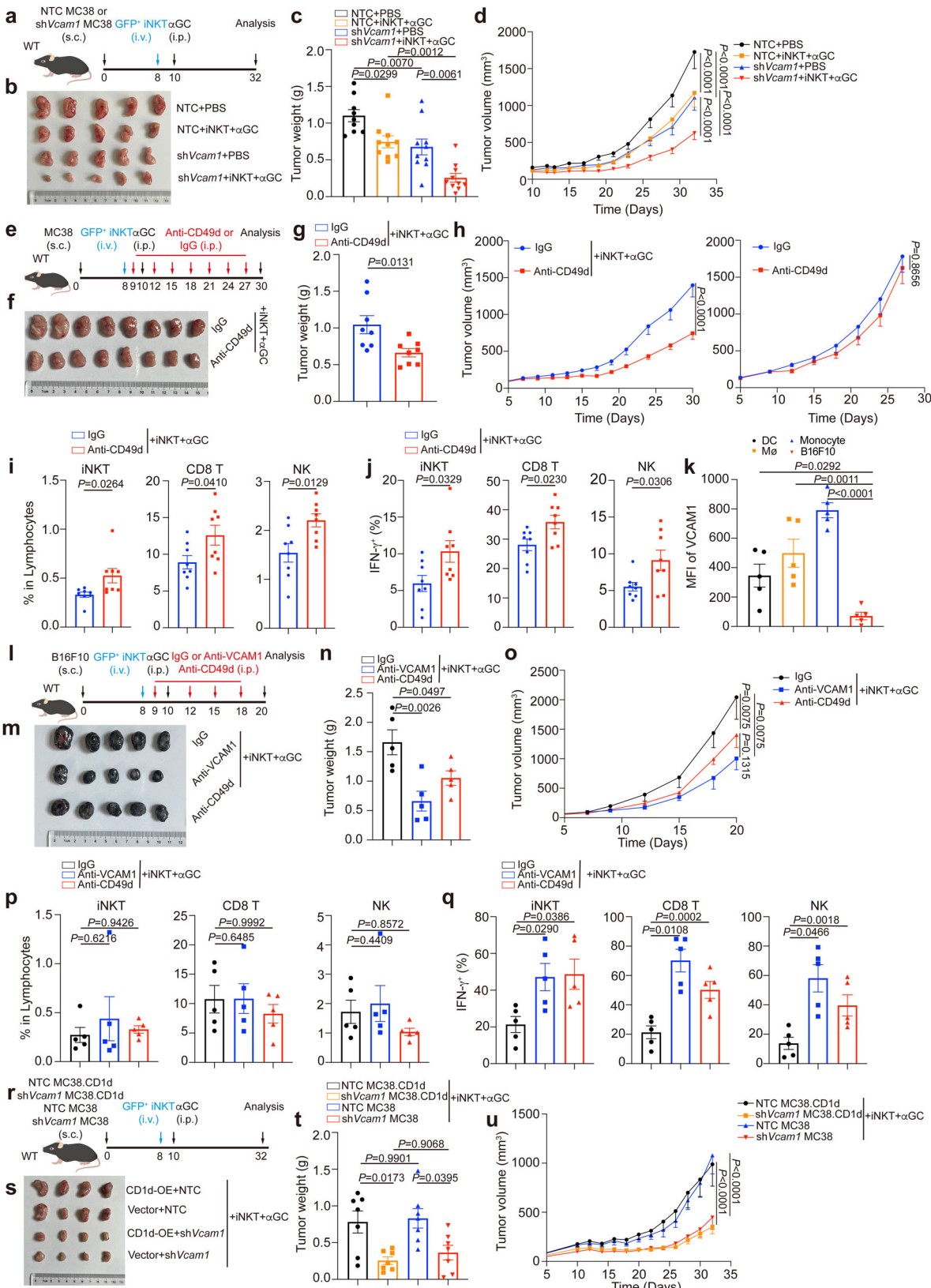

interactions in tumors might favor pro-tumor immune responses in addition to those aforementioned effects. On the other hand, killing tumor macrophages is one aspect of iNKT cell mediated anti-tumor responses[60–62]. It is rational that while iNKT cells kill macrophages via FasL and CD1d molecules[60], macrophages hinder iNKT cell mediated anti-tumor responses via CD1d mediated interactions. Given the

accumulation of macrophages at the boundary of tumors, these cells serve as obstacles to the tumor entry of iNKT cells. On the other hand, chemokines could also make contributions to their tumor entry[10,12]. Although macrophages have been reported to inhibit CD8 T cell motility in tumors[55], those CD1d mediated cell interactions showed no influence on motility of iNKT cells in tumors. Our findings do not

**Fig. 8 | Blockage of VCAM1-CD49d signaling enhances anti-tumor efficacy of iNKT cells. a** Timeline of experimental procedure for (**b**–**d**). **b**–**d** Picture (**b**), weight (**c**), and growth curves (**d**) of *Vcam1* knockdown MC38 tumors and NTC MC38 tumors in WT mice receiving iNKT cells plus αGC or receiving PBS buffer (*n* = 9–10 mice per group). **e** Timeline of experimental procedure for (**f**–**j**). **f**–**g** Picture (**f**) and weight (**g**, *n* = 8 mice per group) of MC38 tumors in WT mice receiving iNKT cells plus αGC combined with anti-CD49d antibody or IgG isotype control antibody. **h** Growth curves of MC38 tumors in WT mice receiving anti-CD49d antibody or IgG isotype control antibody in combination with (*n* = 8 mice per group) or without (*n* = 6 mice per group) iNKT cells plus αGC. **i**, **j** Frequencies (**i**) and IFN-γ production (**j**) of total iNKT cells, CD8 T cells, and NK cells in MC38 tumors described in (**f**, **g**). **k** VCAM1 expression in DCs, macrophages (Mø), monocytes, and tumor cells in

B16F10 tumors from WT mice (*n* = 5 mice). **l** Timeline of experimental procedure for (**m**–**q**). **m**–**q** Picture (**m**), weight (**n**), and growth curves (**o**) of B16F10 tumors in WT mice receiving iNKT cells plus αGC combined with anti-VCAM1 antibody or anti-CD49d antibody or IgG isotype control antibody (*n* = 5 mice per group). **p**, **q** Frequencies (**p**) and IFN-γ production (**q**) of total iNKT cells, CD8 T cells, and NK cells in B16F10 tumors described in (**m**–**o**). **r** Timeline of experimental procedure for (**s**–**u**). NTC, none target control. **s**–**u** Picture (**s**), weight (**t**), and growth curves (**u**) of *Vcam1* knockdown MC38 tumors and NTC MC38 tumors with or without CD1d overexpression in WT mice receiving iNKT cells plus αGC (*n* = 7 mice per group). Data are represented as mean ± SEM, and were analyzed by one-way ANOVA (**c**, **k**, **n**, **p**, **q**, **t**), two-way ANOVA (**d**, **h**, **o**, **u**), and two-tailed unpaired Student's *t*-test (**g**, **i**, **j**). Source data are provided as a Source Data file.

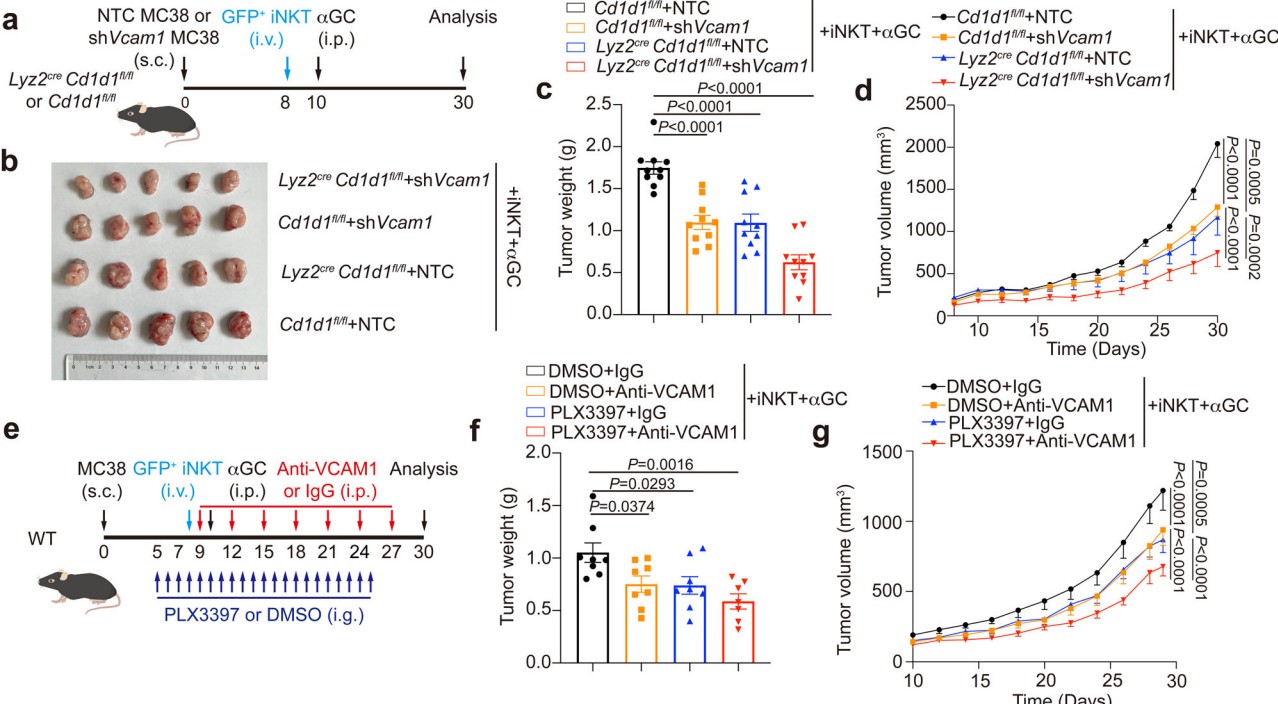

**Fig. 9 | Interfering with macrophage-iNKT cell interactions synergizes with inhibiting VCAM1 signaling to improve efficacy of iNKT cell-based immunotherapy. a** Timeline of experimental procedure for (**b**–**d**). NTC, none target control. **b**–**d** Picture (**b**), weight (**c**), and growth curves (**d**) of *Vcam1* knockdown MC38 tumors and NTC MC38 tumors in *Lyz2^cre Cd1d^fl/fl* mice or *Cd1d^fl/fl* mice receiving iNKT cells plus αGC (*n* = 10 mice per group). **e** Timeline of experimental procedure for (**f**, **g**). **f**, **g** Weight (**f**), and growth curves (**g**) of MC38 tumors in WT

mice receiving iNKT cells plus αGC combined with anti-VCAM1 antibody lone or PLX3397 (40 mg kg⁻¹) alone or anti-VCAM1 antibody plus PLX3397 in combination (*n* = 7–8 mice per group). IgG isotype control antibody and DMSO were used as negative controls. Data are represented as mean ± SEM, and were analyzed by one-way ANOVA (**c**, **f**) and two-way ANOVA (**d**, **g**). Source data are provided as a Source Data file.

exclude the possibility that macrophages might modulate the motility of intratumoral iNKT cells via other molecules[40].

After entering tumors, VCAM1 in tumors, not limited to tumor cells, confines the motility of iNKT cells. In addition to VCAM1 receptor α4β1 integrin, iNKT cells express lymphocyte function-associated antigen 1 (LFA1), a receptor for ICAM1. ICAM1-LFA1 signaling but not VCAM1-CD49d signaling has been previously reported to regulate retention of iNKT cells in spleens and livers[63,64]. In CD8 T cells, ICAM1-LFA1 signaling enhances their tumor accumulation[31,32,65]. Furthermore, adhesion molecules are known to promote immunological synapse formation and signaling transduction[20,42,66]. ICAM1-LFA1 signaling promotes iNKT cell activation as co-stimulatory signaling[63]. Our study suggests that VCAM1-CD49d signaling is not essential for iNKT cell activation in tumors, and is dispensable for tumor cell clearance. In comparison with iNKT cells in those normal tissues, intratumoral iNKT cells expressed higher CD49d but lower LFA1 (Fig. 4b). Additionally, higher VCAM1 but lower ICAM1 was detected in tumor tissues than in normal tissues (Fig. 4a). It is possible that these adhesion

molecules play distinct roles in regulating migration of iNKT cell into/out and inside tumors as well as their activation processes. The different expression pattern of adhesion molecules between tumors and normal tissues might contribute to the tumor exclusion, low motility, and hyporesponsiveness of intratumoral iNKT cells.

Cell motility as well as the formation of immunological synapses are generally controlled by cytoskeleton, particularly actin polymerization[20,29,42]. Indeed, we found reduced expression of CDC42, a small GTPase controlling actin polymerization, in intratumoral iNKT cells, and that was caused by VCAM1-CD49d signaling (Fig. 5f–h). Although antigens presented by macrophages showed no influence on CDC42 expression in tumors, we could not exclude the possibility that other cells might induce reduction of CDC42 in a CD1d-lipid antigen dependent manner according to published data reporting short chain lipid antigen-induced CDC42 reduction[46]. Notably, the VCAM1-CD49d signaling reduced CDC42 expression in CD8 T cells as well (Supplementary Fig. 7b, f), implying impaired motility and activation of these cells in tumors. These results are in line with previous findings that

VCAM1 expression in tumors is associated with tumor evasion and CD8 T cell exclusion[34,38].

Our study proposes a combinational immunotherapy with iNKT cell transfer and VCAM1/CD49d antibody treatment, irrespective the VCAM1 expression and CD1d expression by tumor cells. Notably, interfering with CD1d-mediated macrophage-iNKT cell interactions or depleting macrophages would further enhance the efficacy of this combinational therapy. Given the pro-tumor effect of macrophages, depleting macrophages could be a better choice. Notably, due to the impact of VCAM1 on CD8 T cells and tumor cells, blocking VCAM1-CD49d signaling could also enhance the efficacy of other immunotherapies such as CD8 T cell-based therapies.

## Methods

### Mice
C57BL/6 Wile Type (WT) mice were purchased from the Gempharmatech Co., Ltd. *Cd11c^cre* mice, *Lyz2^cre* mice, *Vα14* Tg *Cxcr6^Gfp* mice, and *Cd1d1^fl/fl* mice have previously been described[41] and were provided by Dr Albert Bendelac. *Ai9* mice were provided by Dr Tian Xue in University of Science and Technology of China[67]. All mice were on the C57BL/6 background. In our study, we used littermate controls and sex-matched mice at 6–12 weeks old, and mice were grouped randomly. Mice were housed under specific pathogen-free conditions with 12 h dark/12 h light cycle. Ambient temperature was maintained at 23 °C with 50% humidity. Mice had ad libitum access to regular mouse chow and acidified water. All animal procedures were approved by the Institutional Animal Care and Use Committee of University of Science and Technology of China, and experiments were performed in accordance with the approved guidelines.

### PBMCs
PBMCs from blood of healthy volunteers were obtained in accordance with protocols approved by the Biomedical Ethics Committee of the Institute of Health and Medicine of Hefei comprehensive national Science Center. Informed consent was obtained from all volunteers.

### Mouse tumor models
For intravital imaging, $1 \times 10^6$ MC38 cells or $1.5 \times 10^6$ *Vcam1* knockdown MC38 cells were subcutaneously injected into the back of mice. In some experiments, $2 \times 10^7$ expanded GFP$^+$ iNKT cells were transferred by paracancerous injection on day 8. To study cytokine responses of iNKT cell in vivo, the tumor-bearing mice were injected intraperitoneally with αGC (Avanti Polar Lipids, 100 μg kg$^{-1}$) for 4 h. To study anti-tumor efficacy of iNKT cells, $1 \times 10^6$ MC38 cells or $2 \times 10^5$ B16F10 cells were subcutaneously injected into mice, and $2 \times 10^7$ expanded iNKT cells were intravenously injected on day 8, then αGC (100 μg kg$^{-1}$) was injected intraperitoneally on day 10. In some experiments, mice were injected intraperitoneally with anti-CD49d antibody (10 mg kg$^{-1}$), anti-VCAM1 antibody (10 mg kg$^{-1}$), or IgG isotype control antibody (10 mg kg$^{-1}$) from day 9 with 2 days intervals. To measure tumor growth, tumor volume was measured from day 8 with 1-day intervals and was calculated as length × width × 0.5 width. For ethical consideration, mice with tumor volume >2000 mm$^3$ were euthanized through carbon dioxide inhalation.

### Intravital imaging of intratumoral iNKT cells
These tumor-bearing mice were implanted with dorsal window chamber on day 10. Tumor-bearing mice were anesthetized by i.p. injecting mebumalnatrium (50 mg kg$^{-1}$) before surgery, and were positioned on a warm plate at 37 °C during surgery. Hair was removed from the back, and the skin on the opposite side of tumor was excised in order to implant dorsal window chamber (APJ Trading Co. Inc) with a 12 mm diameter slide. Mice were injected intraperitoneally with Tolfedine (16 mg kg$^{-1}$) to remit pain, 6 h after surgery. During imaging, mouse was positioned on a warm plate at 37 °C and was anesthetized

by inhalation of 1.5% isoflurane. The time-lapse images were taken with Olympus FVMPE-RS two-photon microscope with a 25× water immersion objective at a frequency of every 2.5 s. The imaging field was about 50–150 μm from tumor edge. The 3D images were taken at z-step of 2.5 μm. Data were analyzed with Imaris software (Version 9.2.0, Bitplane). For cell motility analysis, cells imaged over 120 seconds were included. To show cell trajectory, each cell was tracked for 200 seconds.

### Confocal microscopy
Tissues were collected from MC38-mCherry tumor-bearing mice and fixed with 4% paraformaldehyde for 24 h before they were dehydrated in 30% sucrose for 24 h. Then tissues were embedded in O.C.T medium. Frozen slices (30 μm thick) were blocked with purified anti-CD16/32 (1 μg ml$^{-1}$) in PBS buffer containing 5% FBS and then stained with antibodies at 4 °C for 20–24 h. Images were taken with Zeiss LSM 980 microscope with a 40× oil immersion objective and were analyzed with ImageJ software (Vesion 2.1.0/1.53c, National Institutes of Health). For cell density analysis in tumor slices, 0.04-mm$^2$ regions of interest (ROIs) were chosen at tumor periphery and inside tumor, respectively, based on distance from margin. For cell interaction analysis, 0.046-mm$^2$ regions of interest (ROIs) were chosen in tumor slices.

For in vitro cell tracking, cells were labeled with fluorescent probe CellTrace™ CFSE (Invitrogen, C34554, 5 μM) or CellTrace™ Violet (Invitrogen, C34557, 5 μM) or CellTracker™ Red CMTPX (Invitrogen, C34552, 2 μM). To inhibit CDC42 activity, inhibitor ZCL278 was used at concentration of 100 μM. and the time-lapse images were taken at a frequency of every 1 second. To study the formation of immune synapses, BMDCs-tdTomato or HeLa.hCD1d cells were cultured with αGC (1 μg ml$^{-1}$) in confocal dishes overnight, then expanded mouse GFP$^+$ iNKT cells or sorted human iNKT cells pretreated with fluorescent probe were added into confocal dishes with indicated treatments for 12 h. Images were taken with Zeiss LSM 980 microscope with a 40× oil immersion objective and were analyzed with ImageJ software (Vesion 2.1.0/1.53c, National Institutes of Health).

### Antibodies and flow cytometry
The fluorochrome-labeled or unlabeled antibodies used for flow cytometry analysis including anti-mouse CD16/32 (93), anti-mouse CD45.2 (104), anti-mouse B220 (RA3-6B2), anti-mouse TCRβ (H57.597), anti-mouse CD4 (GK1.5), anti-mouse CD8α (53–6.7), anti-mouse NK1.1 (PK136), anti-mouse CD49d (R1-2), anti-mouse VCAM1 (429), anti-mouse ICAM1 (YN1/1.7.4), anti-mouse LFA1 (H155-78), anti-mouse CD1d (1B1), anti-mouse CD11b (M1/70), anti-mouse CD11c (N418), anti-mouse Ly-6C (HK1.4), anti-mouse IA/IE (M5/114.15.2), anti-mouse CD24 (M1/69), anti-mouse F4/80 (BM8), anti-mouse IFNγ (XMG1.2), anti-mouse CD206 (C068C2) and anti-human CD3 (OKT3), anti-human CD8 (SK1), anti-human CD49d (9F10), anti-human VCAM1 (STA), anti-human IFNγ (4 S.B3), anti-mouse/human CDC42 (EPR15620), anti-mouse/human pSrc (EPR17734), anti-rabbit IgG (Poly4064) were purchased from Biolegend, BDbiosciences or abcam. Purified antibodies including anti-mouse CD49d (PS/2), anti-mouse VCAM1 (M/K-2.7), rat IgG2b (LTF-2) and rat IgG1 (HRPN) were purchased from Bioxcell and used in in vivo experiments. Purified antibodies including anti-mouse CD49d (9C10), anti-mouse VCAM1 (429), rat IgG (RTK2758), anti-mouse CD3 (145-2C11), anti-mouse CD28 (37.51), anti-human VCAM1 (STA), mouse IgG (MOPC-21), anti-human CD3 (OKT3) and anti-human CD28 (CD28.2) were purchased from Biolegend and used in in vitro experiments. In the in vitro blocking experiments, each antibody was used at concentration of 5 μg ml$^{-1}$. mCD1d-PBS57 tetramer and hCD1d-PBS57 tetramer were provided by the NIH Tetramer Core Facility. For apoptosis analysis, cells were stained with Annexin V-FITC/PI Apoptosis Detection Kit (Vazyme, A211-02). Samples were acquired with a BD FACSVerse flow cytometer or BECKMAN CytoFLEX S flow cytometer. Cytometry data were analyzed with FlowJo software (Version 10.5.0, Tree Star, Inc., Ashland, OR).

## Cell expansion and activation

To expand iNKT cells in vitro, splenocytes from $V\alpha14$ Tg $Cxcr6^{Gfp}$ mice were cultured in 1640 medium with 10% FBS and stimulated with αGC (100 ng ml⁻¹) in the presence of IL-2 (100 IU ml⁻¹) for 3 days. Half of medium was changed every 2 days in the presence of IL-2 (100 IU ml⁻¹). On day 10, the purity of iNKT cells was about 85%. Human iNKT cells from PBMCs were expanded in similar conditions with αGC (1 µg ml⁻¹). The purity of human iNKT cells was about 40% on day 10. To activate iNKT cells in vitro, expanded mouse iNKT cells were stimulated with αGC (1 µg ml⁻¹)-pulsed BMDCs for 12 h, and expanded human iNKT cells were stimulated with αGC (1 µg ml⁻¹)-pulsed HeLa.hCD1d cells for 12 h. To evaluate the antigen presenting capability of DCs, intratumoral DCs sorted as CD45.2⁺ Ly-6C⁻ MHCII⁺ F4/80⁻ CD11c⁺ cells and splenic DCs sorted as CD45.2⁺ MHCII⁺ CD11c⁺ cells were respectively co-cultured with sorted hepatic iNKT cells in the presence of αGC (1 µg ml⁻¹) for 12 h. To activate CD8 T cells, CD8 T cell were enriched from mouse splenocytes by Pan T cell isolation kit (Miltenyi Biotec, 130095130) or from human PBMCs by magnetic beads (Miltenyi Biotec, 130090855), and were stimulated with plated-coated anti-CD3 (10 µg ml⁻¹) plus anti-CD28 (10 µg ml⁻¹) antibody for 24 h.

## Cell transfection

To knock down *Vcam1* in MC38 cells, we used lentiviral vector expressing sh*Vcam1* (sequence CCA GAT CCT TAA TAC TGT TTA). For CDC42 overexpression in iNKT cells, expanded mouse iNKT cells on day 3 were transduced with lentivirus (pMDLg-pRRE/pMD2.G/pRSV-REV/pCDH) carry CDC42/EGFP or EGFP as control. To overexpress human CD1d and human VCAM1, cells were transfected with pEGFP-C1-hCD1d and pcDNA3-hVCAM1.

## Primers for qRT-PCR

The following primers were used: *Vcam1*-Forward: GCC CAC TAA ACG CGA AGG T, *Vcam1*-Reverse: ACT GGG TAA ATG TCT GGA GCC, *Icam1*-Forward: GTG ATG CTC AGG TAT CCA TCC A, *Icam1*-Reverse: CAC AGT TCT CAA AGC ACA GCG, *Actb*-Forward: CAT TGC TGA CAG GAT GCA GAA GG, *Actb*-Reverse: TGC TGG AAG GTG GAC AGT GAG G.

## RNA-Seq

cDNA library was sequenced using the Illumina sequencing platform (NovaSeq6000). The size of the library was ~300 bp, and both ends of the library were sequenced to a length of 100 bp. The raw reads were cleaned by removing adapter sequences, short sequences (length < 35 bp), low-quality bases (quality < 20) and ambiguous sequences (i.e., reads with more than two unknown bases 'N'). Hisat2 (version:2.0.4) were used to map the cleaned RNA-seq reads to the mouse mm10 genome with two mismatches, two gaps, and one multihit allowed. After genome mapping, Stringtie (version:1.3.0) was used to quantify gene expression. The gene expression value was normalized by FPKM and adjusted by a geometric algorithm. The RNA-seq was performed by Shanghai Biotechnology Corporation (Shanghai, China).

## Statistics and reproducibility

All experiments were repeated at least twice, and no data were excluded from the analysis. No prior sample size calculation was performed, and sample sizes in the experiments were set based on our previous experience and similar studies in the literature. Prism 8 (Version 8.3.1, GraphPad) was used to perform statistical analyses, Data were represented as mean ± SEM. Two-tailed unpaired Student's *t*-test was used to compare two independent groups with normally distributed samples. One-way analysis of variance (ANOVA) followed by Dunnett's multiple comparisons test was used to compare three or more independent groups. Two-way ANOVA followed by Tukey's multiple comparisons test was used to analyze the tumor growth data. Log-rank test was used to analyze survival curves. Fisher's exact test was used to analyze the significance of relationship between two categorical variables. Pearson's correlation was used to analyze linear association between two variables. *P*-values < 0.05 were considered statistically significant in all experiments.

## Reporting summary

Further information on research design is available in the Nature Portfolio Reporting Summary linked to this article.

## Data availability

RNA-seq datasets reported in this study have been deposited in the National Center for Biotechnology Information Sequence Read Archive under accession number PRJNA946041. Other data supporting our findings are available upon request. Requests for materials should be addressed to L.B. Source data are provided with this paper.

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

## Acknowledgements

We thank NIH Tetramer Core Facility for providing mCD1d-PBS57 tetramer and hCD1d-PBS57 tetramer. We thank Dr. Tian Xue for providing *Ai9* mice, and thank Dr. Albert Bendelac for providing *Cd11c^cre^* mice, *Lyz2^cre^* mice, *Vα14* Tg *cxcr6^Gfp^* mice, and *Cd1d1^fl/fl^* mice. This work was supported by National Key R&D Program of China 2021YFC2300604 (Li.B.), National Natural Science Foundation of China 32325020 (Li.B.), 92254304 (Li.B.), 82071736 (Huimin.Z.), 82101912 (Sicheng.F.), 82372778 (Sicheng.F.), 82202022 (Shuhang.L.), 82371732 (Huimin.Z.), the CAS Project for Young Scientists in Basic Research (YSBR-074, Li.B.), and the Natural Science Foundation of Hefei (2021027, Huimin.Z.).

## Author contributions

Chenxi.T., Sicheng.F., Huimin.Z., and Li.B. conceived the idea and designed the experiments, Chenxi.T., Yu.W., Miya.S., and Yuanyuan.H. performed experiments, Yuwei.Z., Jiaxiang.D., Changfeng.Z., Yuting.C., Jun.P., Shiyu.B., Qielan.W., Sanwei.C., Shuhang.L., Di.X., Rong.L., Yusheng.C., and Yucai.W. provided materials, developed methods, or discussed experiments, Chenxi.T., Huimin.Z., and Li.B. wrote the manuscript. All authors contributed to the article and approved the submitted version.

## Competing interests

The authors declare no competing interests.
