## [Peer Review File · Nature Communications]

Motility and tumor infiltration are key aspects of invariant natural killer T cell anti-tumor functionREVIEWER COMMENTS

Reviewer #1 (Remarks to the Author):

iNKT cells provide anti-tumor activity, but factors that determine their infiltration into tumors are poorly understood. Here, Tian et al., present a study on the infiltration capacity and migratory behavior of iNKT cells in tumors.

Throughout most of the study, the authors use fast growing MC38 cancer cells that are transplanted sub-cutaneously. Using genetic mouse models, the authors can monitor motility of GFP+ iNKT cells using intravital imaging.

The authors show that vast majority of iNKT cells are located at the periphery of tumors, not in the center. The few in the center are less motile.

Regarding antigen recognition, CD1d expression by DC cells ensures activation of iNKT cells.

Alternatively, CD1d expression in macrophages that are also located at the periphery of the tumor inhibits iNKT cells infiltration.

In parallel, the authors study VCAM1 expression levels and impact on iNKT cells. Most specifically, knock down of VCAM1 on tumors cells improves infiltration and motility of iNKT cells.

Mechanistically, VCAM1 signals downregulation of CDC42 in iNKT cells, both in vivo in mouse models, as well as in vitro using human cells. From a therapeutic viewpoint, blocking VCAM1 signals in tumors with antibodies does improve anti-tumor activity of NKT cells.

The downside of the study is its biased approach in following-up on Cd1d expression and VCAM1 signaling that were known suspects. Although the data is new and well presented, it lacks a true discovery of complete unknown biology regarding factors that regulate NKT cell biology. Having said that, the experiments are well conducted, and the manuscript is well structured.

Major comments:

-The predominant cancer model in the study are MC38 cells that generates a fast growing tumor. As a result, analyses are performed on a tumor that is about 10days old, lacking proper time that the tumor cells could educate the TME. A number of key experiments should be performed to validate findings using an alternative cancer model that is analyzed after 2 to 3 months of age to ensure accurate TME composition and activity. In addition, MC38 are colorectal cancer cells, creating the possibility for orthotopic transplantation of the cells in the wall of the caecum to further optimize the type of TME while remaining compatibility with intravital imaging. Alternatively, perhaps spontaneous tumors in mice can also be analyzed to maximize accurate TME composition and actions.

-In figure 2, motility of iNKT cells are analyzed with the presence of normal macrophages or macrophages with depletion of Cd1d. It is not specified which iNKT cells are being tracked. Please divide the data into iNKT cells at the periphery (with and without Cd1d in macrophages) and iNKT cells in the tumor center (with and without Cd1d in macrophages). Based on the limited motility, it seems that only iNKT cells in the center were analyzed, while there are almost no macrophages in the center to begin with.

-There seems to be a discrepancy between the tumor volumes of control mice and the ones treated with iNKT cells + aGC (fig. 6b/c), versus the tumor volumes treated with iNKT cells + aGC in figure 6f/g. In fig6b there seems to be an effect of infusing extra iNKT cells, while this effect seems absent in figure 6f/g. Also, in figure 6f there is no control to monitor tumor volume upon anti-CD49d antibody without the supplementation of iNKT cells.

-Regarding antigen presentation. Is it possible to analyze infiltration patterns upon deletion of Cd1d in DCs, similarly as the studies in figure 2 regarding macrophages? Vice versa, what happens to the activation of iNKT cells upon depletion of Cd1d in macrophages? What happens when Cd1d is depleted in both the macrophages as well as in DC cells (both activation, as well as migration/motility).

-Towards a patient setting, patients with higher VCAM1 tumor levels exhibited lower survival rates. Is it possible to study iNKT infiltrates in these patient samples? Alternatively, correlate VCAM1 expression levels in tumors with iNKT infiltrate/activation using IHC on FFPE material?

-There seems to be a conflicting role of antigen presentation by Cd1d on DCs (activating) versus macrophages (inhibiting infiltration). Can the authors reflect on that in the discussion?

Minor

From studying the abstract, it is not immediately clear what the authors mean with the static or motile regulation paradigm. Please provide brief explanation in abstract for non-expert reader.

Reviewer #2 (Remarks to the Author):

Tian et al investigated the dynamic interactions regulating iNKT cell motility and functions in the tumor microenvironment. This is done by primarily applying intravital microscopy to syngeneic transplantable mouse tumor models injected sc into different recipient mice in which iNKT cell can be tracked by the expression of fluorescent proteins. iNKT cell appear to accumulate at the tumor margin and excluded from the internal area. The few tumor infiltrating iNKT cells are significantly less motile than the majority of cells accumulating at the tumor margin. Similar experiments show also that intratumoral iNKT cells entertain less contacts with CD1d+ DCs compared to the splenic cells, resulting in reduced IFN γ production upon activation by the injection into mice of the strong CD1d-dependent agonist α -GalCer. iNKT cell recognition of CD1d+ myelomonocytic cells promotes their arrest at the tumor margin but has no role on their reduced intratumor motility. Instead, both intratumor infiltration and motility of iNKT cells depends on a VCAM1-dependent interaction with cancer cells, as suggested by the generation of MC38 cancer cells transduced with shVCAM1 RNA. The knockdown of VCAM1 expression by cancer cells recovers also intratumor iNKT cell interactions with DC and their IFN γ production upon α -GalCer injection into tumor bearing mice. Gene expression comparison between iNKT cells from WT or VCAM1 knockdown tumors reveals a differential expression of pathways implicated in cell motility and adhesion, under the control of the Rho GTPase CDC42 signaling axis, which are upregulated in iNKT cells from VCAM1 knockdown tumors supporting their increased motility. Inhibiting the VCAM1-CDC42 signaling axis increases also iNKT cell contacts with CD1d+ DCs and their IFN γ production upon α -GalCer stimulation. The inhibitory VCAM1-CDC42 signaling axis is conserved in human iNKT cells, as wells in mouse and human CD8+ T cells, as suggested by a series of experiments in vitro. Blockade of the VCAM1/CD49d binding by mAb administration in vivo potentiates iNKT cell tumor control irrespective of whether VCAM1 is expressed by cancer cells or by normal infiltrating cells. Inhibiting CD1d-dependent iNKT-macrophages interaction in vivo synergize with VCAM1 knockdown on cancer cells to improve tumor control. It can be concluded that iNKT cells are retained at the tumor margin and that the TME inhibits their motility and functions via CD1d- and VCAM1-dependent interactions, and that VCAM1 may be therapeutically targeted together with macrophages depletion to improve iNKT cell adoptive immunotherapy.

Major comments

1. It would help the interpretation of this study if the authors characterized the population of tumor macrophages that interact with iNKT cells and stop them at the tumor margin;
2. Figure 1 o-s: from these results the authors conclude that intratumoral iNKT cells respond less to α -GalCer injection, compared to splenic iNKT cells, because they are less motile and, therefore, fail to properly scan CD1d+ DCs for antigen encounter. This conclusion needs a further control to rule out that intratumor iNKT cells produce less IFN γ than the splenic ones because there are dysfunctional, rather than less motile. iNKT cells extracted from the spleen or tumor of the same mice should be activated ex vivo and compared for IFN γ secretion;
3. Figure 5 s-v. It would be interesting to show human iNKT cell adhesion, activation and IFN γ production upon interaction in vitro with a human cancer cell line co-expressing both CD1d and VCAM1, +/- blocking with mAbs directed against either molecule;
4. Previous work by Song L, doi: 10.1172/JCI37869; Cortesi F, doi: 10.1016/j.celrep.2018.02.058; Lanakiram NB, doi: 10.1111/imm.12746 show that iNKT cells kill and/or modulate CD1d-expressing tumor macrophages and that this is a critical aspect of their anti-tumor effector response. Authors should quote these studies in the manuscript and discuss these results in light

of their seemingly different findings in terms of iNKT cell-tumor macrophage outcome.

Minor comments

1. Please show primary data for Figure 1 s-r in supplements
2. The shRNA methods are apparently lacking;
3. Figure 7 d, g: the tumor progression curves need statistics.

Reviewer #3 (Remarks to the Author):

Invariant natural killer T (iNKT) cells have direct and indirect anti-tumor effects, and their dysfunction limit their anti-tumor efficacy. Several studies have explored the modulation of iNKT cells in a relatively static paradigm. However, the factors controlling motility of iNKT cells and their contribution to impaired iNKT cells' function remain to be explored.

In the current manuscript Tian et al. described the mechanisms underlying the restricted motility of iNKT cells in tumors. Through multiple experimental models and technological strategies, they identified that VCAM1 expressed by tumor cells downregulates Cdc42 gene expression in iNKT cells through interaction with integrin receptors containing CD49d. Such signaling limits macrophage motility in tumors, which in turn makes them fail to scan and respond to antigen and thus reduces the anti-tumor efficacy. In addition, the authors also pay parallel attention to macrophages as interacting with iNKT cells via CD1d, hindering the latter's activation and anti-tumor function. They further verified these findings is conserved between mouse and human.

Overall, their findings are novel and meaningful. They provided a new strategy to kill tumor via a combinational immunotherapy with iNKT cells transfer and VCAM1/CD49d antibody treatment. Experiments were well designed and performed to high standards. However, there are a few questions should be addressed.

Major concern:

1. The authors described changes in the motility of iNKT cells, as well as changes in tumor progression. How can high- motility iNKTs better deal with cancer cells beyond by releasing more IFN- γ ? It is possible to determine whether more apoptotic tumor cells are associated with iNKT by TUNLE staining.
2. The authors found knockdown of Vcam1 in MC38-mCherry tumor cells improved infiltration of GFP + transferred iNKT cells into tumors (Fig. 3e-f), but they also found knockdown of Vcam1 alone, without iNKT cell transfer plus α GC injection, impaired tumor growth (in line 233 to 235, Extended Data Fig. 3b). In knockdown of Vcam1, they need to figure out the contribution of autonomous MC38 tumor cell death or immune cells based on iNKT cells.
3. Vcam1 knockdown MC38 tumors indicated improved CDC42 signaling in iNKT cells, in which signal pathway? does overexpression of CDC42 in iNKT cells or other agonists of increasing iNKT cells motility also inhibit the growth of tumor? The authors need more evidences to support the viewpoint "our data indicate that tumor VCAM1 inhibits iNKT cell motility and activation via reducing CDC42 expression." (in line 186 and 187).

Minor concern:

1. The authors claimed, "the densities of iNKT cells inside tumors (about 100 μ m from the margin) were much lower than they were at tumor periphery" (in line 87-88). However, in Supplementary Movie 1, there are similar number of iNKT cells.
2. There are obvious less iNKT cells in tumor in Fig. 1q. Thus, we can expect less iNKT-DC cluster in field in Fig.1r. The statistical analysis needs to be normalized by cell number to confirm "iNKT cells formed clusters with DCs more efficiently in spleens than in tumors in vivo" (in line 110-111).

3. Why anti-VCAM1 or anti-CD49d antibody treatment could significantly enhance anti-tumor efficacy of iNKT cells in B16F10 tumor models, with low/no level of Vcam1 expressing In B16F10 tumor cells? (Fig. 6k) Would Vcam1 antibody treatment influence apoptosis in B16F10 tumor cells? (Fig. 6l-n) Do iNKT cells have the same great difference motility like MC38 tumor model?

4. The authors described the interaction between iNKT cells and DCs and tumor VCAM1. What will happen when use VCAM1 antibody in Cd11ccre Cd1d1fl/fl mice? According to Fig. 3m, DCs were also influenced in Vcam1 knockdown MC38-mcherry tumors. The authors should revise the manuscript to make this point clearer, at least discuss it in the discussion.

5. The authors should provide supplementary movie to show the interaction between iNKT cells with DCs or macrophage.

REVIEWER COMMENTS

Reviewer #1 (Remarks to the Author):

iNKT cells provide anti-tumor activity, but factors that determine their infiltration into tumors are poorly understood. Here, Tian et al., present a study on the infiltration capacity and migratory behavior of iNKT cells in tumors.

Throughout most of the study, the authors use fast growing MC38 cancer cells that are transplanted sub-cutaneously. Using genetic mouse models, the authors can monitor motility of GFP+ iNKT cells using intravital imaging.

The authors show that vast majority of iNKT cells are located at the periphery of tumors, not in the center. The few in the center are less motile.

Regarding antigen recognition, CD1d expression by DC cells ensures activation of iNKT cells. Alternatively, CD1d expression in macrophages that are also located at the periphery of the tumor inhibits iNKT cells infiltration.

In parallel, the authors study VCAM1 expression levels and impact on iNKT cells. Most specifically, knock down of VCAM1 on tumors cells improves infiltration and motility of iNKT cells. Mechanistically, VCAM1 signals downregulation of CDC42 in iNKT cells, both in vivo in mouse models, as well as in vitro using human cells. From a therapeutic viewpoint, blocking VCAM1 signals in tumors with antibodies does improve anti-tumor activity of NKT cells.

The downside of the study is its biased approach in following-up on Cd1d expression and VCAM1 signaling that were known suspects. Although the data is new and well presented, it lacks a true discovery of complete unknown biology regarding factors that regulate NKT cell biology. Having said that, the experiments are well conducted, and the manuscript is well structured.

Major comments:

-The predominant cancer model in the study are MC38 cells that generates a fast growing tumor. As a result, analyses are performed on a tumor that is about 10days old, lacking proper time that the tumor cells could educate the TME. A number of key experiments should be performed to validate findings using an alternative cancer model that is analyzed after 2 to 3 months of age to ensure accurate TME composition and activity. In addition, MC38 are colorectal cancer cells, creating the possibility for orthotopic transplantation of the cells in the wall of the caecum to further optimize the type of TME while remaining compatibility with intravital imaging. Alternatively, perhaps spontaneous tumors in mice can also be analyzed to maximize accurate TME composition and actions.

Response: Thanks for raising this issue. We understand the reviewer's concerns. We could not use 2-3 months old tumors according to the animal ethical guidelines. According to the reviewer's suggestion, we imaged the motility of iNKT cells in MC38 tumors over one month old, and observed impaired motility of intratumoral iNKT cells in comparison with iNKT cells at tumor periphery. Additionally, in those experiments evaluating anti-tumor efficacy, tumors were 4-5 weeks old. At these timepoints, the TME was generated considering the impaired behavior and function of intratumoral

immune cells. On the other hand, generating spontaneous tumor models takes long time, so we performed similar experiments with B16F10 tumor models, in addition to MC38 tumor models, and obtained similar results. Briefly, iNKT cells in B16F10 tumors showed impaired motility as shown in Fig. 1s-v, and blocking VCAM1-CD49d signaling in B16F10 tumor-bearing mice enhanced the intratumoral function and anti-tumor effects of iNKT cells (Fig. 8m-q). These results suggest general inhibitory effects of VCAM1-CD49d signaling on iNKT cells in tumors, irrespective of tumor types.

-In figure 2, motility of iNKT cells are analyzed with the presence of normal macrophages or macrophages with depletion of Cd1d. It is not specified which iNKT cells are being tracked. Please divide the data into iNKT cells at the periphery (with and without Cd1d in macrophages) and iNKT cells in the tumor center (with and without Cd1d in macrophages). Based on the limited motility, it seems that only iNKT cells in the center were analyzed, while there are almost no macrophages in the center to begin with.

Response : We understand the reviewer's concern. Although macrophages were enriched at the tumor boundary, there were lots of macrophages inside tumor, as shown in Fig. 3a and published studies^{1,2}. To investigate the influence of macrophage CD1d on intratumoral iNKT cell behavior, we imaged iNKT cells inside tumors (with and without Cd1d in macrophages) as shown in Fig. 3g-k. We found that CD1d expression in macrophages showed no influence on iNKT cell motility in tumors.

-There seems to be a discrepancy between the tumor volumes of control mice and the ones treated with iNKT cells + aGC (fig. 6b/c), versus the tumor volumes treated with iNKT cells + aGC in figure 6f/g. In fig6b there seems to be an effect of infusing extra iNKT cells, while this effect seems absent in figure 6f/g. Also, in figure 6f there is no control to monitor tumor volume upon anti-CD49d antibody without the supplementation of iNKT cells.

Response: Thanks for bringing up this issue. Experiments related to Fig. 6b/c and Fig. 6f/g were performed at different times, therefore we could not compare them directly. In Fig. 6f/g, both the isotype control treated mice and anti-CD49d antibody treated mice received iNKT cell transfer plus α GC, and the data proved elevated anti-tumor efficacy of iNKT cells by anti-CD49d antibody treatment. According to reviewer's suggestion, we treated MC38 tumor-bearing mice with isotype control or anti-CD49d antibody, without the supplementation of iNKT cells plus α GC. We found that anti-CD49d antibody alone failed to significantly inhibit tumor growth. These data are now shown in Fig. 8h. Consistently, although knockdown of *Vcam1* increased apoptosis of MC38 tumor cells, anti-VCAM1 antibody or anti-CD49d antibody treatment showed no influence on their apoptosis. These data are now shown in Supp. Fig. 4f. These results support that blocking VCAM1-CD49d signaling enhances anti-tumor effects via modulating iNKT cell function. Indeed, increased IFN γ production was detected in intratumoral iNKT cells after blocking VCAM1-CD49d signaling, as shown in Fig. 8j.

-Regarding antigen presentation. Is it possible to analyze infiltration patterns upon deletion of Cd1d in DCs, similarly as the studies in figure 2 regarding macrophages? Vice versa, what happens to the activation of iNKT cells upon depletion of Cd1d in macrophages? What

happens when Cd1d is depleted in both the macrophages as well as in DC cells (both activation, as well as migration/motility).

Response: Thanks for bringing up these issues. To address those questions, we used *lyz2 cre Cd1d^{fl/fl}* mice and *Cd11c cre Cd1d^{fl/fl}* mice. In *lyz2 cre Cd1d^{fl/fl}* mice, CD1d was deleted in macrophages but not in DCs. In *Cd11c cre Cd1d^{fl/fl}* mice, CD1d was deleted in both macrophages and in DCs. Therefore, the differences between these two mice were due to the deletion of CD1d in DCs. Given the dramatically reduced α -Galactosylceramide (α GC)-induced IFN- γ production in splenic iNKT cells and intratumoral iNKT cells from *Cd11c cre Cd1d^{fl/fl}* mice but enhanced IFN- γ production in those cells from *Lyz2 cre Cd1d^{fl/fl}* mice, DCs were predominant antigen presenting cells activating and inducing Th1 anti-tumor response of iNKT cells in tumors as well as in spleens *in vivo*^{3,4}. These data are now shown in Fig. 2a-c, and are in line with DCs' properties including high capacity of antigen presentation and promoting Th1 response of iNKT cells^{3, 4}. Additionally, we found that deletion of CD1d in macrophages increased tumor infiltration of iNKT cells but showed no influence on their motility in tumors. Notably, deletion of CD1d in macrophages increased DC-iNKT interactions and enhanced frequencies of IFN γ ⁺ iNKT cells in tumors after α GC injection. These data are now shown in Fig. 3f and Fig. 2c, and indicate that CD1d mediated macrophage-iNKT interactions interfere with CD1d mediated DC-iNKT interactions and Th1 response in iNKT cells. The increased iNKT-DC interactions caused by deleting CD1d in macrophages are likely due to absence of competition between macrophages and DCs to interact with iNKT cells. Despite the priority of DCs in antigen presentation, macrophages might interfere with iNKT-DC interactions considering their large cell numbers.

According to reviewer's suggestion, we also imaged iNKT cell motility in tumors of *Cd11c cre Cd1d^{fl/fl}* mice. In line with published studies that antigen stimulation reduces iNKT cell motility^{5, 6}, impaired antigen presentation in *Cd11c cre Cd1d^{fl/fl}* mice led to elevated motility of iNKT cells in tumors. Therefore, both tumor VCAM1-CD49d signaling and antigen recognition on DCs impairs iNKT cell motility, but they lead to different outcomes. One interferes with antigen scanning and iNKT cell activation, whereas the other leads to activation of iNKT cells. These *Cd11c cre Cd1d^{fl/fl}* mice also had increased tumor infiltration of iNKT cells. However, in consistence with the failure of iNKT cell activation in *Cd11c cre Cd1d^{fl/fl}* mice, these mice favored tumor growth in comparison with control mice. These data are shown below. In the manuscript, we focus on factors influencing iNKT cell infiltration and antigen scanning and mainly discuss the roles of macrophages.

-Towards a patient setting, patients with higher VCAM1 tumor levels exhibited lower survival rates. Is it possible to study iNKT infiltrates in these patient samples? Alternatively, correlate VCAM1 expression levels in tumors with iNKT infiltrate/activation using IHC on FFPE material?
Response: We appreciate the reviewer's suggestion. With data in TCGA-LGG, we found that VCAM1 level was negatively correlated with level of iNKT cell signature genes, *TRAV24* and *ZBTB16*, indicating poor iNKT cell infiltration in VCAM1 high expression LGG patients. This data is now shown in Fig. 7b.

-There seems to be a conflicting role of antigen presentation by Cd1d on DCs (activating) versus macrophages (inhibiting infiltration). Can the authors reflect on that in the discussion?
Response: We appreciate the reviewer's suggestion. We now discuss this in the discussion section in manuscript (shown below).

In addition to high capacity in antigen presentation, DCs are known to favor Th1 response of iNKT cells, which is important for anti-tumor immunity^{3, 4}. Therefore, CD1d expression in DCs is essential for iNKT cell-mediated anti-tumor immune responses. Conversely, CD1d expression in macrophages inhibits tumor infiltration of iNKT cells and interferes with iNKT-DC interactions as well as Th1 response in tumors. *In vivo*, CD1d expressing antigen presenting cells could compete to interact with iNKT cells, despite the priority of DCs. Published studies have demonstrated that different antigen presenting cells interact with iNKT cells and lead to distinct immune responses⁷. Notably, M2 macrophages are known to shape iNKT cell mediated responses toward Th2^{7, 8}. Given the M2 phenotype of tumor macrophages, we could not exclude the possibility that macrophage-iNKT cell interactions in tumors might favor pro-tumor immune responses in addition to those aforementioned effects.

Minor

From studying the abstract, it is not immediately clear what the authors mean with the static or motile regulation paradigm. Please provide brief explanation in abstract for non-expert reader.

Response: We appreciate the reviewer's suggestion. We add brief explanation in abstract (shown below).

Current mechanistic studies mainly focus on their dysfunctional status before or after activation, a relatively static regulation paradigm, whereas the motile regulation

paradigm related to antigen scanning process remains unexplored.

Reviewer #2 (Remarks to the Author):

Tian et al investigated the dynamic interactions regulating iNKT cell motility and functions in the tumor microenvironment. This is done by primarily applying intravital microscopy to syngeneic transplantable mouse tumor models injected sc into different recipient mice in which iNKT cell can be tracked by the expression of fluorescent proteins. iNKT cell appear to accumulate at the tumor margin and excluded from the internal area. The few tumor infiltrating iNKT cells are significantly less motile than the majority of cells accumulating at the tumor margin. Similar experiments show also that intratumoral iNKT cells entertain less contacts with CD1d+ DCs compared to the splenic cells, resulting in reduced IFN γ production upon activation by the injection into mice of the strong CD1d-dependent agonist α -GalCer. iNKT cell recognition of CD1d+ myelomonocytic cells promotes their arrest at the tumor margin but has no role on their reduced intratumor motility. Instead, both intratumor infiltration and motility of iNKT cells depends on a VCAM1-dependent interaction with cancer cells, as suggested by the generation of MC38 cancer cells transduced with shVCAM1 RNA. The knockdown of VACM1 expression by cancer cells recovers also intratumor iNKT cell interactions with DC and their IFN γ production upon α -GalCer injection into tumor bearing mice. Gene expression comparison between iNKT cells from WT or VACM1 knockdown tumors reveals a differential expression of pathways implicated in cell motility and adhesion, under the control of the Rho GTPase CDC42 signaling axis, which are upregulated in iNKT cells from VCAM1 knockdown tumors supporting their increased motility. Inhibiting the VCAM1-CDC42 signaling axis increases also iNKT cell contacts with CD1d+ DCs and their IFN γ production upon α -GalCer stimulation. The inhibitory VCAM1-CDC42 signaling axis is conserved in human iNKT cells, as wells in mouse and human CD8+ T cells, as suggested by a series of experiments in vitro. Blockade of the VCAM1/CD49d binding by mAb administration in vivo potentiates iNKT cell tumor control irrespective of whether VCAM1 is expressed by cancer cells or by normal infiltrating cells. Inhibiting CD1d-dependent iNKT-macrophages interaction in vivo synergize with VCAM1 knockdown on cancer cells to improve tumor control. It can be concluded that iNKT cells are retained at the tumor margin and that the TME inhibits their motility and functions via CD1d- and VCAM1-dependent interactions, and that VCAM1 may be therapeutically targeted together with macrophages depletion to improve iNKT cell adoptive immunotherapy.

Major comments

1. It would help the interpretation of this study if the authors characterized the population of tumor macrophages that interact with iNKT cells and stop them at the tumor margin;

Response: We appreciate the reviewer's suggestion. We characterized the macrophages at tumor boundary and inside tumor, and found that these macrophages including those interacting with iNKT cells were F4/80⁺ CD11c⁻ CD206⁺, exhibiting M2 phenotypes. These data are now shown in Suppl. Fig. 3.

2. Figure 1 o-s: from these results the authors conclude that intratumoral iNKT cells respond

less to a-GalCer injection, compared to splenic iNKT cells, because they are less motile and, therefore, fail to properly scan CD1d+ DCs for antigen encounter. This conclusion needs a further control to rule out that intratumor iNKT cells produce less IFN γ than the splenic ones because there are dysfunctional, rather than less motile. iNKT cells extracted from the spleen or tumor of the same mice should be activated *ex vivo* and compared for IFN γ secretion;

Response: Thanks for bringing up this issue. Our data indicate that tumor VCAM1 impairs iNKT cell motility via reducing CDC42 expression. Inhibiting CDC42 activity reduced iNKT cell motility and impaired BMDC-iNKT interactions as well as iNKT cell activation. Via overexpressing CDC42 in iNKT cells, we restored motility of iNKT cells *in vitro* in the presence of VCAM1 expressing tumor cells (Fig. 6l-n). Additionally, we confirmed that, *in vivo*, CDC42 overexpression led to enhanced tumor infiltration, IFN γ production, and anti-tumor efficacy of iNKT cells. These results are now shown in Fig. 6o-s, and support the importance of intratumoral motility on iNKT cell function. Notably, we do not exclude that other factors lead to dysfunction of iNKT cells in tumors. As we discussed in the manuscript, both static regulation paradigm related to dysfunctional status and motile regulation paradigm related to antigen scanning process contribute to impaired function of iNKT cells in tumors.

3. Figure 5 s-v. It would be interesting to show human iNKT cell adhesion, activation and IFN γ production upon interaction *in vitro* with a human cancer cell line co-expressing both CD1d and VCAM1, +/- blocking with mAbs directed against either molecule;

Response: We understand reviewer's concern. *In vivo*, DCs predominantly present antigens to iNKT cells, despite expression of CD1d by other immune and non-immune cells, and that is due to the high antigen presenting capacity of DCs. Even tumor cells express CD1d molecule, they could not present lipid antigens as efficiently as DCs do. MDA-MB-231.hVCAM1 tumor cells did not express CD1d. In Fig. 7s-w, we cocultured human iNKT cell, MDA-MB-231.hVCAM1 tumor cells, and Hela.hCD1d cells mimicking DCs, and treated them with isotype control or anti-VCAM1 antibody. We found that anti-VCAM1 antibody increased the motility and IFN γ production of human iNKT cells as well as their interaction with Hela.hCD1d cells. To investigate the influence of CD1d expression in tumor cells, we generated MDA-MB-231.hVCAM1.hCD1d tumor cells which expressed median level of CD1d, and those CD1d high expressing Hela.hCD1d cells were used to mimic DCs with high antigen presentation capacity. We cocultured human iNKT cell, MDA-MB-231.hVCAM1.hCD1d tumor cells, and Hela.hCD1d cells, and treated them with isotype control or anti-VCAM1 antibody. We found that anti-VCAM1 antibody increased the motility and IFN γ production of iNKT cells as well as their interaction with Hela.hCD1d cells. These data are now shown in Supp. Fig.6a-e, and indicate that blocking VCAM1 signaling enhances iNKT cell antigen scanning and activation irrespective of CD1d expression in tumor cells. Additionally, this conclusion is further supported by *in vivo* experiment using mouse tumor models (with or without CD1d expression in tumor cells), as shown in Fig. 8r-u.

4. Previous work by Song L, doi: 10.1172/JCI37869; Cortesi F, doi:

10.1016/j.celrep.2018.02.058; Lanakiram NB, doi: 10.1111/imm.12746 show that iNKT cells kill and/or modulate CD1d-expressing tumor macrophages and that this is a critical aspect of their anti-tumor effector response. Authors should quote these studies in the manuscript and discuss these results in light of their seemingly different findings in terms of iNKT cell-tumor macrophage outcome.

Response: We appreciate the reviewer's suggestion. We quoted these studies in the manuscript and discuss these results in discussion section.

On the other hand, killing tumor macrophages is one aspect of iNKT cell mediated anti-tumor responses^{9, 10, 11}. It is rational that while iNKT cells kill macrophages via FasL and CD1d molecules⁹, macrophages hinder iNKT cell mediated anti-tumor responses via CD1d mediated interactions.

Minor comments

1. Please show primary data for Figure 1 s-r in supplements

Response: We appreciate the reviewer's suggestion. We now show primary data for Fig. 2e-f in Suppl. Fig. 2d.

2. The shRNA methods are apparently lacking;

Response: Thanks for raising this issue. We add the shRNA information in methods section.

3. Figure 7 d, g: the tumor progression curves need statistics.

Response: Thanks for raising this issue. We add the statistics in Fig. 9d, g

.

Reviewer #3 (Remarks to the Author):

Invariant natural killer T (iNKT) cells have direct and indirect anti-tumor effects, and their dysfunction limit their anti-tumor efficacy. Several studies have explored the modulation of iNKT cells in a relatively static paradigm. However, the factors controlling motility of iNKT cells and their contribution to impaired iNKT cells' function remain to be explored.

In the current manuscript Tian et al. described the mechanisms underlying the restricted motility of iNKT cells in tumors. Through multiple experimental models and technological strategies, they identified that VCAM1 expressed by tumor cells downregulates Cdc42 gene expression in iNKT cells through interaction with integrin receptors containing CD49d. Such signaling limits macrophage motility in tumors, which in turn makes them fail to scan and respond to antigen and thus reduces the anti-tumor efficacy. In addition, the authors also pay parallel attention to macrophages as interacting with iNKT cells via CD1d, hindering the latter's activation and anti-tumor function. They further verified these findings is conserved between mouse and human.

Overall, their findings are novel and meaningful. They provided a new strategy to kill tumor via a combinational immunotherapy with iNKT cells transfer and VCAM1/CD49d antibody

treatment. Experiments were well designed and performed to high standards. However, there are a few questions should be addressed.

Major concern:

1. The authors described changes in the motility of iNKT cells, as well as changes in tumor progression. How can high- motility iNKTs better deal with cancer cells beyond by releasing more IFN- γ ? It is possible to determine whether more apoptotic tumor cells are associated with iNKT by TUNLE staining.

Response: Thanks for raising this issue. It has been reported that iNKT cells promote NK and CD8 T cell mediated tumor clearance, in addition to their direct killing⁴. In our study, we found that both knockdown of *Vcam1* and blocking VCAM1-CD49d signaling enhanced activation of NK cells and CD8 T cells, in addition to iNKT cells. All of these cells could contribute to enhanced tumor clearance. According to reviewer's suggestion, we performed TUNLE staining and found more apoptotic tumor cells around iNKT cells in MC38 tumors from mice received iNKT cells plus α GC and anti-VCAM1 antibody, in comparison with those mice received iNKT cells plus α GC. These data are now shown below.

2. The authors found knockdown of *Vcam1* in MC38-mCherry tumor cells improved infiltration of GFP + transferred iNKT cells into tumors (Fig. 3e-f), but they also found knockdown of *Vcam1* alone, without iNKT cell transfer plus α GC injection, impaired tumor growth (in line 233 to 235, Extended Data Fig. 3b). In knockdown of *Vcam1*, they need to figure out the contribution of autonomous MC38 tumor cell death or immune cells based on iNKT cells.

Response: We understand the reviewer's concern. Although knockdown of *Vcam1* increased MC38 tumor cell apoptosis, blocking VCAM1-CD49d signaling via anti-CD49d antibody or anti-VCAM1 antibody alone showed no influence on MC38 tumor cell apoptosis *in vitro* or on tumor growth *in vivo*, as shown in Suppl. Fig. 4f. and Fig. 8h. Notably, anti-CD49d antibody combined with iNKT cell plus α GC showed better anti-tumor efficacy than iNKT cell plus α GC alone in MC38 tumor models. Additionally, anti-CD49d antibody and anti-VCAM1 antibody showed no influence on B16F10 tumor cell apoptosis *in vitro*, but enhanced iNKT cell mediated anti-tumor efficacy *in vivo*. In all the cases, inhibiting VCAM1-CD49d signaling *in vivo* increased activation of iNKT cells, NK cells, and CD8 T cells. These results confirm that inhibiting VCAM1-CD49d signaling enhances anti-tumor responses via promoting iNKT cell-mediated immune responses. In *Vcam1* knockdown tumors, we did not exclude the possible contribution of autonomous cell death.

3. Vcam1 knockdown MC38 tumors indicated improved CDC42 signaling in iNKT cells, in which signal pathway? does overexpression of CDC42 in iNKT cells or other agonists of increasing iNKT cells motility also inhibit the growth of tumor? The authors need more evidences to support the viewpoint "our data indicate that tumor VCAM1 inhibits iNKT cell motility and activation via reducing CDC42 expression." (in line 186 and 187).

Response: Thanks for the great suggestions and we performed several experiments to address the reviewer's questions. We found that tumor VCAM1 reduced phosphorylation of Src in iNKT cells, and inhibiting Src activity in iNKT cells led to reduction of CDC42 expression. Our data indicate that tumor VCAM1 reduces CDC42 expression in iNKT cells via Src signal pathway. Additionally, to further support that CDC42 reduction in iNKT cells led to impaired motility and anti-tumor function, we overexpressed CDC42 in iNKT cells. We found that, in the presence of VCAM1 expressing tumor cells, overexpression of CDC42 restored motility of iNKT cells *in vitro*. Additionally, we confirmed that, *in vivo*, CDC42 overexpression led to enhanced tumor infiltration, IFN γ production, and anti-tumor efficacy of iNKT cells. These data are now shown in Fig. 6l-s and Suppl. Fig. 5a-b.

Minor concern:

1. The authors claimed, "the densities of iNKT cells inside tumors (about 100 μ m from the margin) were much lower than they were at tumor periphery" (in line 87-88). However, in Supplementary Movie 1, there are similar number of iNKT cells.

Response: Thanks for bringing up this issue. As we showed in Fig. 1a-b, densities of iNKT cells in tumors were lower than they were at tumor periphery. We have changed the movie which could better represent our data. The data are now shown in Fig. 2e.

2. There are obvious less iNKT cells in tumor in Fig. 1q. Thus, we can expect less iNKT-DC cluster in field in Fig.1r. The statistical analysis needs to be normalized by cell number to confirm "iNKT cells formed clusters with DCs more efficiently in spleens than in tumors *in vivo*" (in line 110-111).

Response: Thanks for the reviewer's suggestion. We have normalized the number of iNKT-DC clusters by iNKT cell number.

3. Why anti-VCAM1 or anti-CD49d antibody treatment could significantly enhance anti-tumor efficacy of iNKT cells in B16F10 tumor models, with low/no level of Vcam1 expressing In B16F10 tumor cells? (Fig. 6k) Would Vcam1 antibody treatment influence apoptosis in B16F10 tumor cells? (Fig. 6l-n) Do iNKT cells have the same great difference motility like MC38 tumor model?

Response: We understand the reviewer's concern. As we showed in Fig. 8k, other cells such as monocytes in B16F10 tumors expressed VCAM1 as well, which could interfere with iNKT cell function via VCAM1-CD49d signaling instead of B16F10 cells. According to reviewer's suggestion, we imaged motility of iNKT cells in B16F10 tumors and observed impaired motility as found in MC38 tumors. These data are now

shown in Fig. 1s-v. Additionally, anti-CD49d antibody and anti-VCAM1 antibody showed no influence on B16F10 tumor cell apoptosis *in vitro*, but enhanced iNKT cell mediated anti-tumor efficacy *in vivo*. These data are now shown in Suppl. Fig. 4g and Fig. 8m-q. These results suggest that VCAM1 molecule in tumors, not limited in tumor cells, inhibits anti-tumor responses of iNKT cells. We have explained it in manuscript.

4. The authors described the interaction between iNKT cells and DCs and tumor VCAM1. What will happen when use VCAM1 antibody in *Cd11c^{cre} Cd1d1^{fl/fl}* mice? According to Fig. 3m, DCs were also influenced in *Vcam1* knockdown MC38-mcherry tumors. The authors should revise the manuscript to make this point clearer, at least discuss it in the discussion. Response: We apologize for having not explained clearly. As shown in Fig. 2a-c, DCs were predominant antigen presenting cells activating and inducing Th1 anti-tumor response of iNKT cells in tumors. In *Cd11c^{cre} Cd1d^{fl/fl}* mice, iNKT cells failed to be efficiently activated due to the absence of antigen presentation. For this reason, these mice failed to control tumor growth efficiently after iNKT cell transfer plus α GC injection (data shown below). In wide type mice, interfering with VCAM1-CD49d signaling led to increased iNKT-DC interaction and enhanced anti-tumor efficacy, but in *Cd11c^{cre} Cd1d^{fl/fl}* mice we do not expect similar effects. According to reviewer's suggestion, we have discussed the role of DCs in discussion section.

5. The authors should provide supplementary movie to show the interaction between iNKT cells with DCs or macrophage.

Response: We understand the reviewer's concern. However, generating reporter mice to perform these experiments would take long time. To study those cell interactions, we used *CD1d* conditional KO mice and imaged cell interactions in tumor slices, and these results support our conclusions. Now, we have shown the iNKT cell function in tumors of *Lyz2^{cre} Cd1d^{fl/fl}* mice as well as in tumors of *Cd11c^{cre} Cd1d^{fl/fl}* mice, and have shown the iNKT-DC interactions and iNKT-macrophage interactions in tumors of *Lyz2^{cre} Cd1d^{fl/fl}* mice and control mice.

Given the dramatically reduced α GC-induced IFN- γ production in splenic iNKT cells and intratumoral iNKT cells from *Cd11c^{cre} Cd1d^{fl/fl}* mice but enhanced IFN- γ production in those cells from *Lyz2^{cre} Cd1d^{fl/fl}* mice (Fig. 2b-c), DCs were predominant antigen presenting cells activating and inducing Th1 anti-tumor response of iNKT cells in tumors as well as in spleens *in vivo*. Notably, deletion of *CD1d* in macrophages increased DC-iNKT interactions and enhanced frequencies of IFN γ ⁺ iNKT cells in tumors after α GC injection, despite the reduced iNKT-macrophage interactions. These data are now shown in Fig. 3c-f and Fig. 2c, and indicate that *CD1d* mediated macrophage-iNKT interactions interfere with *CD1d* mediated DC-iNKT interactions

and Th1 response in iNKT cells. The increased iNKT-DC interactions caused by deleting CD1d in macrophages are likely due to absence of competition between macrophages and DCs to interact with iNKT cells. Despite the priority of DCs in antigen presentation, macrophages might interfere with iNKT-DC interactions considering their large cell numbers.

References

1. Christofides, A. *et al.* The complex role of tumor-infiltrating macrophages. *Nat Immunol* **23**, 1148-1156 (2022).
2. Bied, M., Ho, W.W., Ginhoux, F. & Bleriot, C. Roles of macrophages in tumor development: a spatiotemporal perspective. *Cell Mol Immunol* **20**, 983-992 (2023).
3. Fujii, S.I. & Shimizu, K. Immune Networks and Therapeutic Targeting of iNKT Cells in Cancer. *Trends Immunol* **40**, 984-997 (2019).
4. Brennan, P.J., Brigl, M. & Brenner, M.B. Invariant natural killer T cells: an innate activation scheme linked to diverse effector functions. *Nat Rev Immunol* **13**, 101-117 (2013).
5. Wong, C.H. & Kubers, P. Imaging natural killer T cells in action. *Immunol Cell Biol* **91**, 304-310 (2013).
6. Liew, P.X. & Kubers, P. Intravital imaging - dynamic insights into natural killer T cell biology. *Front Immunol* **6**, 240 (2015).
7. Bai, L. *et al.* Distinct APCs explain the cytokine bias of alpha-galactosylceramide variants in vivo. *J Immunol* **188**, 3053-3061 (2012).
8. Cruz, M.S., Loureiro, J.P., Oliveira, M.J. & Macedo, M.F. The iNKT Cell-Macrophage Axis in Homeostasis and Disease. *Int J Mol Sci* **23** (2022).
9. Cortesi, F. *et al.* Bimodal CD40/Fas-Dependent Crosstalk between iNKT Cells and Tumor-Associated Macrophages Impairs Prostate Cancer Progression. *Cell Rep* **22**, 3006-3020 (2018).
10. Janakiram, N.B. *et al.* Loss of natural killer T cells promotes pancreatic cancer in LSL-Kras(G12D/+) mice. *Immunology* **152**, 36-51 (2017).
11. Song, L. *et al.* Valpha24-invariant NKT cells mediate antitumor activity via killing of tumor-associated macrophages. *J Clin Invest* **119**, 1524-1536 (2009).

REVIEWERS' COMMENTS

Reviewer #1 (Remarks to the Author):

The authors adequately addressed my major concerns, including the addition of new experimental data. In particular a later stage analysis, new tumor model and mouse genetics to discriminate between CD1d function on macrophages or DCs)

Reviewer #2 (Remarks to the Author):

The authors have addressed all my concerns, including doing new ad hoc experiments. In my opinion, the study is much improved and acceptable for publication.

Reviewer #3 (Remarks to the Author):

Authors have addressed all my questions. This manuscripts now is ready to get published.

REVIEWER COMMENTS

Reviewer #1 (Remarks to the Author):

The authors adequately addressed my major concerns, including the addition of new experimental data. In particular a later stage analysis, new tumor model and mouse genetics to discriminate between CD1d function on macrophages or DCs)

Response: We gratefully appreciate for their reviewer's support.

Reviewer #2 (Remarks to the Author):

The authors have addressed all my concerns, including doing new ad hoc experiments. In my opinion, the study is much improved and acceptable for publication.

Response: We gratefully appreciate for their reviewer's support.

Reviewer #3 (Remarks to the Author):

Authors have addressed all my questions. This manuscripts now is ready to get published.

Response: We gratefully appreciate for their reviewer's support.